# Iceberg melting substantially modifies oceanic heat flux towards a major Greenlandic tidewater glacier

B. J. Davison [1✉], T. R. Cowton [1], F. R. Cottier [2,3] & A. J. Sole [4]

Fjord dynamics influence oceanic heat flux to the Greenland ice sheet. Submarine iceberg melting releases large volumes of freshwater within Greenland's fjords, yet its impact on fjord dynamics remains unclear. We modify an ocean model to simulate submarine iceberg melting in Sermilik Fjord, east Greenland. Here we find that submarine iceberg melting cools and freshens the fjord by up to ~5 °C and 0.7 psu in the upper 100-200 m. The release of freshwater from icebergs drives an overturning circulation, resulting in a ~10% increase in net up-fjord heat flux. In addition, we find that submarine iceberg melting accounts for over 95% of heat used for ice melt in Sermilik Fjord. Our results highlight the substantial impact that icebergs have on the dynamics of a major Greenlandic fjord, demonstrating the importance of including related processes in studies that seek to quantify interactions between the ice sheet and the ocean.

[1] Department of Geography and Sustainable Development, University of St Andrews, St Andrews, UK. [2] Scottish Association for Marine Science, Scottish Marine Institute, Oban, UK. [3] Department of Arctic and Marine Biology, UiT The Arctic University of Norway, Tromsø, Norway. [4] Department of Geography, University of Sheffield, Sheffield, UK. ✉email: bd41@st-andrews.ac.uk

The dynamics of Greenland's glacial fjords control the transport of oceanic heat to Greenland's tidewater glaciers, with potentially important implications for ice-sheet stability[1] and global sea level[2]. For example, the rapid retreat of many of Greenland's tidewater glaciers during the early 2000s has been attributed to increased oceanic forcing[3–6], due to both ocean warming and invigorated fjord circulation resulting from enhanced ice-sheet runoff[7,8]. The resultant increase in glacier submarine melt rates[9] may have led to greater undercutting of glacier calving fronts[10,11] and an increase in glacier calving rates[12–14]. Understanding the controls on oceanic heat flux to tidewater glacier calving fronts is therefore essential if we are to reliably predict the influence of the ocean on Greenland's tidewater glaciers in a changing climate.

Oceanic heat flux towards Greenland's tidewater glaciers depends on the rate of fjord-shelf exchange, the properties of the ocean waters entering the fjords, and how those waters are modified during fjord transit prior to interacting with tidewater glaciers[15]. Off the coast of Greenland, two water masses provide the principal oceanic input to Greenland's fjords[16]. Cooler, fresher water of Polar origin ('Polar Water') is found in the upper 100–200 m of the water column, and is typically underlain by warmer, more saline water of Atlantic origin ('Atlantic Water'). Where deep glacially eroded troughs extend to the continental slope, they allow the passage of Atlantic Water and both water masses can access Greenland's fjords[7]. A range of processes force these water masses into the fjords, including (for example) tidal mixing[17], barrier winds[18,19] and buoyancy-driven circulation controlled by ice-sheet runoff[8]. Once in the fjord, the temperature-salinity structure set by these water masses is modified by a range of processes. During the spring and summer months, freshwater inputs from sea ice melt, terrestrial snow melt and precipitation cool and freshen surface and near-surface waters[20]. In addition, the discharge of fresh glacial runoff from tidewater glacier grounding lines (henceforth runoff) drives buoyant plumes[21], which entrain and transport relatively warm Atlantic Water towards the glaciers and subsequently towards the fjord surface[15,22–24]. This drives an outflowing current of glacially modified water in the upper layers of the fjord and a compensatory inflowing current over a broad depth range below, typically between the Atlantic Water-Polar Water interface and the sill depth[25–27].

Icebergs are a major component of Greenland's glacial fjords. Frontal ablation (iceberg calving plus submarine melting of glacier termini) at tidewater glaciers represents ~30–50% of the freshwater export from the ice sheet into the ocean, of which calving of icebergs is the larger component[28]. Icebergs melt partially or entirely whilst transiting glacial fjords[29–31], thereby providing a heat sink and fresh water source that is distributed horizontally and vertically throughout the fjord. The resultant freshwater flux comprises a key component of the freshwater budget of iceberg-congested fjords[29,32,33].

Despite the prevalence of icebergs in many of Greenland's fjords and the substantial release of freshwater from them, the impact of their melting on fjord water properties, fjord circulation and therefore oceanic heat flux towards tidewater glaciers remains largely unknown. This study is motivated by the hypothesis that icebergs substantially modify fjord water properties, which may in turn affect fjord circulation and the oceanic forcing of tidewater glaciers. This is informed by summertime observations of water properties along Sermilik Fjord—one of the largest and most thoroughly surveyed fjords in East Greenland (Fig. 1)—which show a marked up-fjord decrease in both temperature and salinity[26,32,34]. This cooling and freshening is confined primarily to the upper few hundred metres of the water column and can be of high magnitude (~5 °C and 0.5 psu). The temperature-salinity

signature of this along-fjord trend is consistent with ice melting in ocean water, rather than of runoff[23,32,34]. Furthermore, inferred ice-melt volume within Sermilik Fjord is an order of magnitude larger than that expected from melting of glacier termini alone[23,32], which suggests that there is a large additional input of meltwater, most likely from submarine iceberg melting. This inference is supported by a small number of hydrographic surveys conducted near icebergs, which identified areas of upwelling and cooling[35–38]. These lines of evidence suggest that submarine iceberg melting may be responsible for considerable modification of fjord water properties, which may in turn affect fjord circulation.

In this study, we quantify the impact of submarine iceberg melting on fjord circulation, fjord water properties and up-fjord oceanic heat flux during summer in Sermilik Fjord. To achieve this, we adapt a numerical ocean model to include a representation of submarine iceberg-ocean interaction, and compare model output to identical simulations without icebergs. We generate a high-fidelity model domain representative of Sermilik Fjord, with realistic bathymetry[39] (Fig. 1a) and an observation-based iceberg distribution[40] (Fig. 1b,c; Supplementary Fig. 1; Methods). Individual cuboidal icebergs are roughly oriented with the fjord long-axis and are represented as a set of horizontal and vertical ice faces, with dimensions based on observed iceberg aspect ratios[41] and relationships between iceberg volume and submerged surface area[40,42] (Methods).

Fjords are dynamic systems, with changes in circulation, iceberg cover and hydrographic conditions occurring over timescales of days to years. It would be computationally intractable (and scientifically confusing) to simulate the full gamut of possible conditions in Sermilik Fjord. Consequently, we concentrate on the summertime regime during which most observations are acquired and when the circulation is thought to be dominated by subglacial runoff[8], and use a single snapshot-in-time of iceberg cover and distribution[40]. It is therefore the interaction between submarine iceberg melting and the circulation driven by glacial runoff that we focus on here. More specifically, we quantify the impact that submarine iceberg melting has on summertime fjord water properties, circulation and therefore oceanic heat flux towards Helheim Glacier, which is the largest glacier terminating in Sermilik Fjord (Fig. 1a), and the second-largest glacier in Greenland in terms of ice discharge[43]. In doing so, we provide the first assessment of how submarine iceberg melting can affect oceanic heat flux towards a major Greenlandic tidewater glacier during summer.

Our simulation design is as follows (see Methods for detail). Each simulation was run for 100 days, reaching a quasi-steady-state (with domain-averaged kinetic energy changing by <3% over the final 10 model days; Supplementary Fig. 2). Runoff, iceberg cover (Fig. 1b, c) and boundary conditions (Fig. 1d) were kept constant throughout each simulation; the diagnostics presented below are averages of the final 10 model days. We used six different runoff values, ranging from 0 to 2000 m³ s⁻¹, to represent the range of summertime runoff discharge into Sermilik Fjord[44]. Simulations with these runoff values were repeated using two subglacial drainage configurations that are broadly representative of 'channelised' and 'distributed' subglacial drainage (Methods). Our primary simulations used melt rate parameter values commonly used in the literature[8] (Methods; Supplementary Table 1); however, recent observations[45,46] suggest that these 'standard' values may underestimate glacier submarine melt rates. We therefore repeated each simulation using the 'adjusted' parameter values (Supplementary Table 1) suggested by Jackson et al.[45]. For clarity, we focus primarily on two simulations using our 'channelised' runoff configuration and the standard parameter values: a 'no-runoff forcing' scenario, which allows us to quantify iceberg-

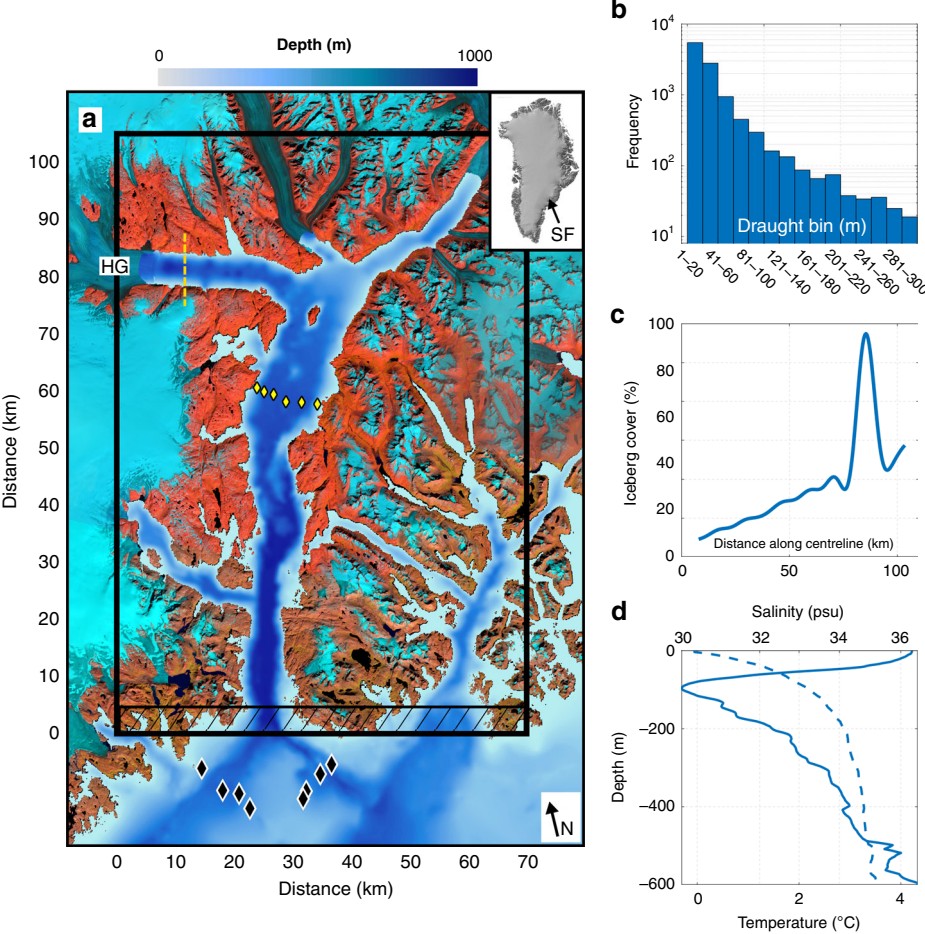

**Fig. 1 Study area and model domain. a** BedMachine v3[39] bathymetry of Sermilik Fjord, with the inset showing its location in Greenland. The inset was created using the Greenland Ice Mapping Project digital elevation model[78]. The background image is a mosaic of five Landsat 8 false-colour composites (bands 6, 5 and 4) acquired on 12 July, 21, 23, 28 and 15 August 2017. HG = Helheim Glacier. **b** Iceberg draught-frequency distribution used in the primary simulations, based on observations by Sulak et al.[40]. **c** The percent of the fjord surface in plan-view covered by icebergs. **d** Temperature (solid line) and salinity (dashed line) boundary conditions. The black box in **a** denotes the limit of the model domain. The black hatching in **a** denotes the relaxation zone (Methods). The black diamonds in **a** indicate locations of conductivity-temperature-depth casts used for boundary conditions (**d**) and the yellow diamonds are conductivity-temperature-depth casts used in Fig. 7. The yellow dashed line in **a** is the flux gate used in our heat flux calculations (Methods).

fjord interaction in isolation, and a 'summer runoff forcing' scenario, in which 1200 m³ s⁻¹ runoff emerges from Helheim Glacier. All simulations are compared to identical simulations without icebergs.

## Results

**Submarine iceberg melt**. Our domain-averaged submarine iceberg melt rates (using standard parameter values) are in the range ~0.09–0.20 m d⁻¹ (Fig. 2a), but grid cell-average melt rates reach 1.34 m d⁻¹ in certain locations and domain-averaged melt rates with the adjusted parameters range from 0.25–0.57 m d⁻¹. Regardless of melt rate parameter values, domain-average melt rates generally increase with runoff raised to the power 0.09–0.12 (Fig. 2a), due to the relatively fast and warm plume outflow increasing heat transfer to the icebergs (particularly those in the vicinity of glacier fronts). For a given increase in runoff, domain-averaged melt rates increase more in the 'distributed' scenario compared to the 'channelised' scenario, because the plume outflow affects a greater proportion of the fjord in the former.

The total freshwater flux released by submarine iceberg melting ranges from ~400 to ~2830 m³ s⁻¹, depending on runoff, subglacial drainage system structure and melt rate parameter values (Fig. 2a). Freshwater release of this magnitude constitutes

an important component of the fjord freshwater budget. For example, in our summer runoff forcing scenario with standard parameter values, the freshwater flux from iceberg melting was 77.4% of the average runoff entering the fjord during July 1990–2012 (Methods). In general, iceberg freshwater production decreased below ~100 m (Fig. 2b). Nevertheless, iceberg melting below the Atlantic Water-Polar Water interface (defined as the 27.3 potential density isopycnal[22]) still contributed a substantial 39.6 ± 11.5% to the total iceberg freshwater flux (Fig. 2b).

Submarine iceberg melt rates are generally greatest at the head of the fjord (Fig. 3a), where iceberg draughts are deepest (Supplementary Fig. 1). Consistent with this spatial distribution, submarine iceberg melt rates generally increase with iceberg draught (Fig. 3b). However, the increase in melt rate with iceberg draught is not linear. There is considerable variability in the melt rates of small icebergs (those with draughts <140 m). Since these comparatively small icebergs do not penetrate below the pycnocline, many of them reside in relatively cool near-surface waters and so melt more slowly. They only interact with relatively warm, fast-flowing currents (and therefore melt more rapidly) where they are exposed to plume outflow in the vicinity of glacier fronts. In contrast, icebergs with greater draughts are more consistently exposed (at least partially) to warmer waters at depth,

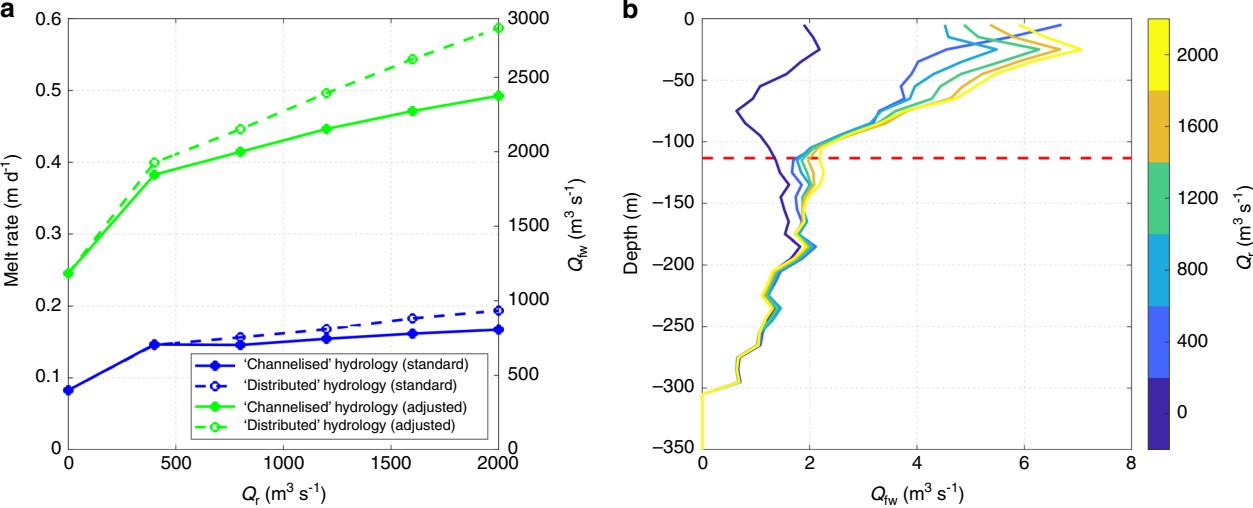

**Fig. 2 Iceberg melt rates and freshwater flux. a** Relationship between runoff ($Q_r$) and domain-averaged submarine iceberg melt rate and total iceberg freshwater flux ($Q_{fw}$). **b** Horizontally averaged iceberg freshwater flux profile, coloured by runoff, for the 'channelised' runoff configuration and standard melt rate parameter values. The dashed horizontal red line in **b** donates the 27.3 potential density contour, approximating the depth of the interface between Polar Water and Atlantic Water. In **a**, the blue lines indicate simulations with standard melt rate parameter values, whilst the green lines indicate simulations with adjusted melt rate parameter values (Methods; Supplementary Table 1). The solid lines represent our 'channelised' drainage scenario and the dashed lines represent our 'distributed' drainage scenario.

and so generally melt faster and with less variability than small icebergs. In terms of melt rates, this results in two populations of icebergs: 'small' icebergs with generally lower, but highly spatially variable melt rates, and 'large' icebergs, with consistently higher melt rates (Fig. 3b). The higher melt rates of deeply-draughted icebergs mean that the freshwater flux from individual icebergs increases slightly super-linearly with submerged iceberg surface area (assuming the same aspect ratio; Fig. 3c; Supplementary Fig. 3), potentially providing a simple method to estimate iceberg freshwater flux from satellite-derived iceberg areas and volumes.

**Impact on Fjord properties and circulation**. In our no-runoff forcing scenario, the freshwater flux released by submarine iceberg melting is capable of generating a weak circulation (Fig. 4). This circulation is characterised by generally down-fjord currents between the fjord surface and ~180 m depth (0.01 m s⁻¹ on average), with peak speeds of ~0.02 m s⁻¹ at ~130 m, driven by the release of freshwater from icebergs. This down-fjord current is underlain by a weaker (~0.006 m s⁻¹) but thicker up-fjord current, peaking at ~270 m and unidentifiable below ~500 m. This weak but broad up-fjord current compensates (in terms of volume) for the fjord water entrained in the relatively fresh and cold iceberg melt-driven outflow above (Fig. 4c).

In the summer runoff forcing scenario, the iceberg melt-driven circulation and the runoff-driven circulation augment one another when their respective currents are aligned and compete when they are not (Fig. 5). For example, some currents in the upper 180 m are slowed by 10–40% because icebergs act as a physical barrier to water flow (Methods) and because in some places the iceberg melt-driven circulation opposes the stronger runoff-driven circulation (Fig. 5a, b). (For example, shallow up-fjord currents, which can be formed in simulations where the plumes reach neutral buoyancy below the surface, are opposed by iceberg-melt-driven currents at the same depth). In contrast, in the mélange and near the fjord walls in the upper 180 m, the iceberg melt-driven circulation augments the runoff-driven circulation (Fig. 5a, b). Throughout the fjord as a whole, up-fjord currents in the 190–500 m depth range are over 30% faster than the equivalent no-iceberg scenario (Fig. 5c, d). It is worth

noting that there are feedbacks here that are difficult to disentangle: the runoff-driven circulation itself increases submarine iceberg melt rates, leading to a stronger iceberg melt-driven circulation, which may in turn impede or augment the runoff-driven circulation depending on the respective directions of each circulatory regime. The overall effect of these modifications to the circulation is to reduce across-fjord heterogeneity in velocity and to increase fjord water export by ~10% in the summer runoff forcing scenario, compared to the equivalent no-iceberg simulation.

Submarine iceberg melting also causes marked changes in the temperature and salinity of the fjord. Without runoff, we simulate iceberg-induced cooling and freshening of up to 5 °C and 0.7 psu throughout the upper ~100 m, though the changes are most pronounced near the fjord head and surface (Fig. 4b). With the addition of runoff, the invigorated circulation results in more uniform iceberg-induced cooling of ~1 °C and freshening of ~0.1 psu above the Atlantic Water-Polar Water interface (Fig. 4e). These modifications produce along-fjord gradients in temperature and salinity in the upper ~100 m of the water column, with water properties migrating towards cooler and less saline conditions with increasing distance from the mouth, except where there is warm plume outflow (Fig. 4). The freshening and cooling have compensating effects on the fjord water density. Thus, the net effect on density from iceberg melt is small (typically much <1% change in any location, compared to the no-iceberg scenario) and the resulting lateral density gradients within the fjord are weak.

**Impact on oceanic heat flux**. We find that the up-fjord oceanic heat flux (Methods) across a flux gate placed within the ice mélange close to the fjord head increases with runoff raised to the power 0.11–0.52, with the exponent dependent on the spatial pattern of runoff efflux across each glacier's grounding line and melt rate parameter values (Fig. 6a). The form of the relationship between runoff and up-fjord heat flux is similar in simulation suites with and without icebergs, indicating that the addition of icebergs in the model does not fundamentally modify the response of these fjords to runoff. In our 'distributed'

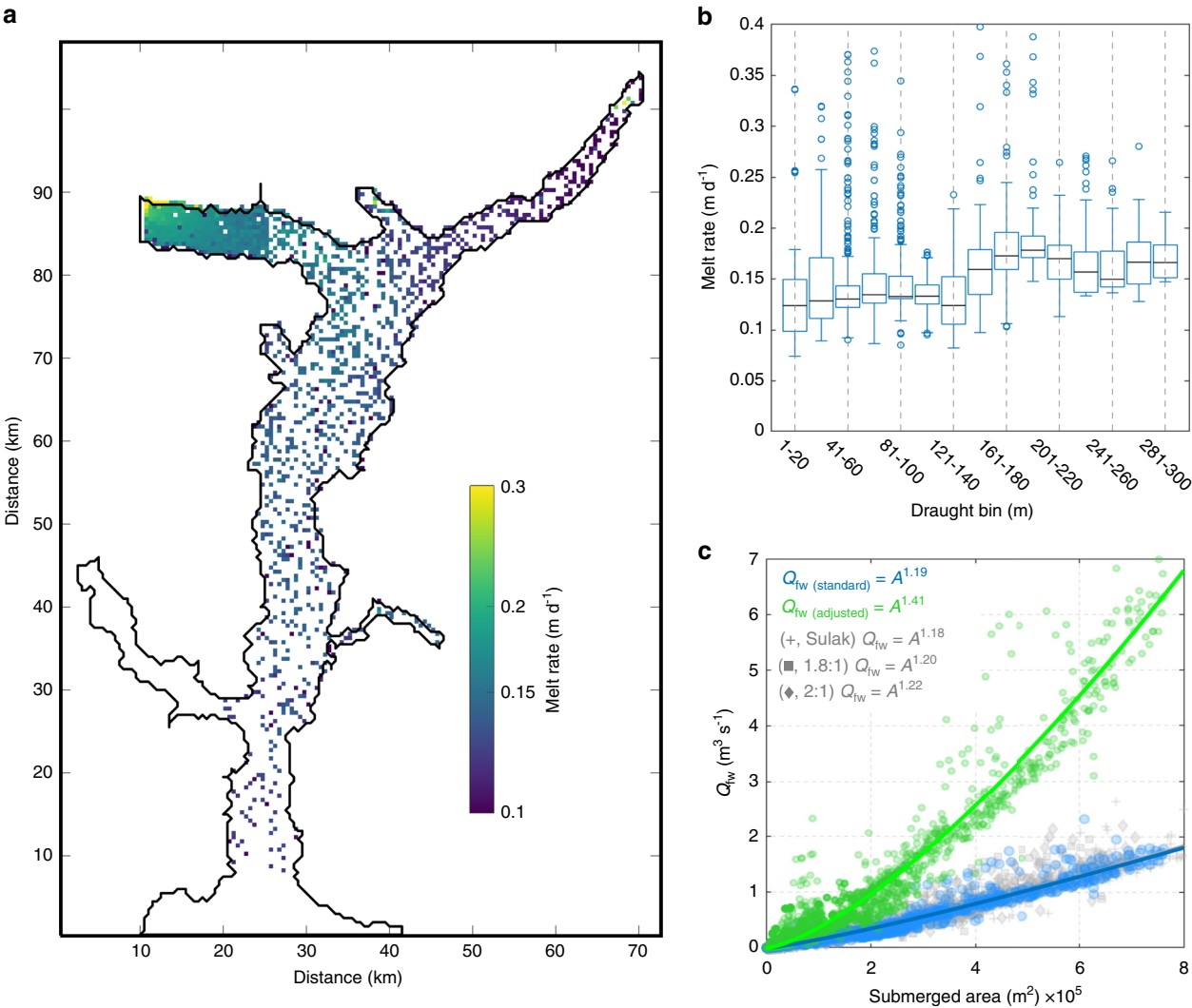

**Fig. 3 Relationship between iceberg geometry and melt rates in our summer runoff forcing scenario. a** Vertically-averaged submarine iceberg melt rates (note log scale). **b** Box plots showing vertically-averaged submarine melt rates of individual icebergs compared to iceberg draught. Each plot shows the median value (black line), interquartile range (box) and all other data (whiskers) except outliers (circles), with outliers defined as data points more than 2.7 standard deviations from the median value. Vertical dashed grey lines in **b** are for visual aid. **c** Scatter plot of submerged iceberg area and freshwater flux ($Q_{fw}$) from individual icebergs, with power-law fits (solid lines) and associated equations (where $A$ is the submerged iceberg surface area). The blue and green coloured circles in **c** correspond to simulations with the standard and adjusted (Methods) melt rate parameter values respectively. The grey crosses, squares and diamonds in **c** correspond to simulations using the volume-area scaling of Sulak et al.[40] (Methods), and length to keel depth ratios of 1.8:1 and 2:1, respectively, all with the standard melt rate parameter values. Axis distances in **a** correspond to those in Fig. 1a.

drainage scenario, the volume transport across the flux gate increases because of more efficient entrainment of ambient waters into glacial plumes per unit volume of runoff (as in Cowton et al.[8]), due to the sub-linear relationship between entrainment of ambient waters into plumes and runoff[47]. This results in an increase in up-fjord oceanic heat flux by 45.5 ± 23.3% compared to the 'channelised' scenario (with the precise value depending on runoff), and a larger exponent in the power-law relationship between runoff and up-fjord oceanic heat flux (Fig. 6a).

The effect of icebergs on up-fjord oceanic heat flux is complex. Excluding the no-runoff forcing scenario, the inclusion of icebergs results in an overall increase in up-fjord oceanic heat flux by 9.3 ± 4.5% with the standard melt rate parameter values, but this increases to 38.4 ± 10.8% with the adjusted values. This overall increase in up-fjord heat flux is most pronounced with low values of runoff and with 'channelised' hydrology (Fig. 6a), when

the iceberg melt-driven circulation is a relatively more important driver of fjord circulation. This depth-averaged effect, however, masks significant variation with depth (Fig. 6b). We simulate a 44.9% reduction in up-fjord oceanic heat flux in the upper 20 m, but a 71.1% increase in up-fjord heat flux in the 30–500 m depth range in our summer runoff forcing scenario with the standard parameter values. This vertical pattern arises because of cooling and weakening of up-fjord surface currents found in the mélange (Fig. 4f) and the strengthening of up-fjord currents below (Fig. 5c, d). In the no-runoff forcing simulations, the circulation is entirely driven by submarine iceberg melting, resulting in an overall >100% increase in up-fjord oceanic heat flux compared to corresponding no-iceberg simulations. We note that the magnitude of the net effect presented here will likely be sensitive to the location of our flux gate and to the boundary conditions used, and so we place more emphasis on the vertical pattern of up-fjord heat flux change.

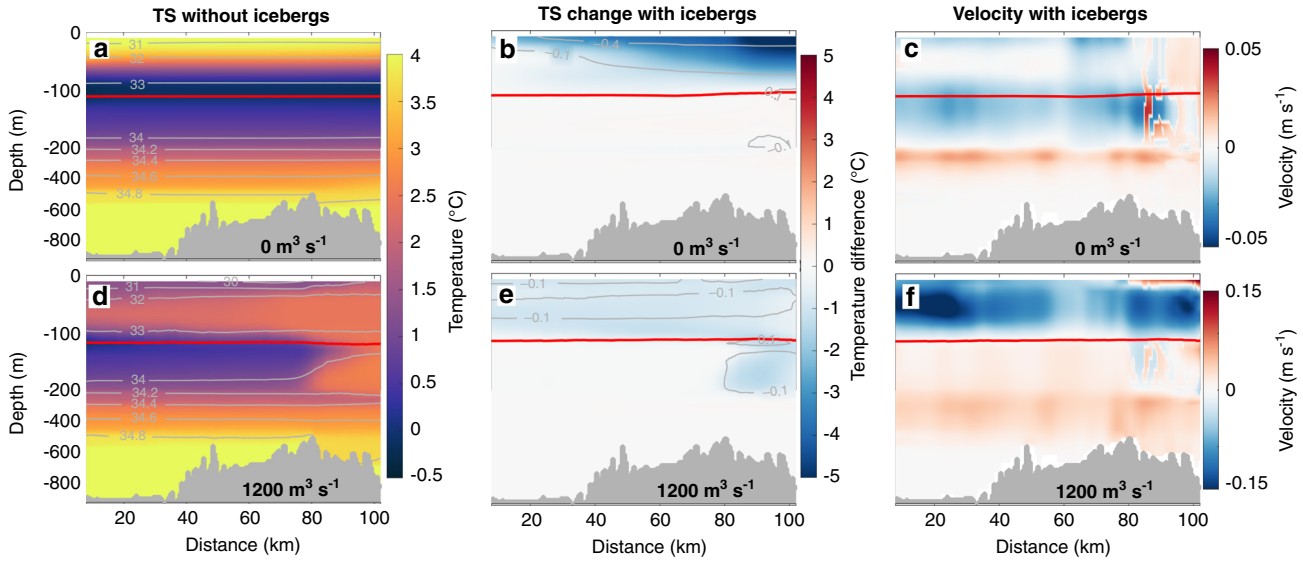

**Fig. 4 Along-fjord transects of water properties and circulation.** Effect of submarine iceberg melting on fjord water properties and circulation using 'channelised' subglacial hydrology and **a–c** no runoff or **d–f** 1200 m³ s⁻¹ runoff. All panels show transects along the fjord centreline, with the ocean boundary on the left. **a, d** Centreline temperature and salinity in the no-iceberg simulations. **b, e** The difference in centreline temperature and salinity in the corresponding iceberg simulations. **c, f** Centreline current velocity in the corresponding iceberg simulations, with positive values indicating up-fjord currents. The red line in each panel denotes the 27.3 potential density contour, approximating the interface between Polar Water and Atlantic Water. Note the y-axis scale is stretched in the upper 200 m. x-axis distances correspond to those in Fig. 1a.

## Discussion

Domain-averaged submarine iceberg melt rates range from 0.09 to 0.57 m d⁻¹ (Fig. 2a), but melt rates in certain grid cells reach 1.34 m d⁻¹. Relatively few estimates of submarine iceberg melt rates and freshwater fluxes are available for comparison. Summertime submarine melt rate estimates for individual large icebergs in the Sermilik Fjord mélange of ~0.39 ± 0.18 m d⁻¹ [48] and ~0.21 ± 0.15 m d⁻¹ [29], based on changes in iceberg freeboard, are similar to the upper-end of our estimates for deeply-draughted icebergs with the standard melt rate parameter values (Fig. 3b). As a further point of comparison, Moon et al.[33] found vertically-averaged melt rates of ~0.36 ± 0.17 m d⁻¹ for individual icebergs and local melt rates of up to ~1 m d⁻¹. Our modelled fjord-wide iceberg freshwater fluxes (400–930 m³ s⁻¹ or 1180–2830 m³ s⁻¹ with the adjusted parameter values) are comparable to previous estimates based on scaling up modelled[33] or inferred[31] melt rates for individual icebergs using observed iceberg size-frequency distributions. Therefore, although we expect our modelled melt rates to be somewhat conservative (due to excluding some melt processes), these comparisons give us confidence that our model is realistically capturing iceberg melting within the fjord. We note that modelled iceberg melt rates are sensitive to a range of uncertain or temporally variable parameter values, including currents driven by melt-driven convection (Methods; Supplementary Figs. 4 and 5), iceberg concentration (Supplementary Fig. 6), maximum iceberg draught (Supplementary Fig. 7) and iceberg aspect ratio (Supplementary Fig. 8), but emphasise that our simulated melt rates are in broad agreement with previous estimates, regardless of these parameter values.

We find that submarine iceberg melting causes substantial cooling and freshening of the upper 100–200 m of Sermilik Fjord (Fig. 4b, e). The impact on water column temperature and salinity increases towards the fjord head, where iceberg concentrations are greatest, resulting in along-fjord gradients in temperature, salinity and density. A similar pattern of up-fjord cooling and freshening is also apparent in the available observations[26,34,49,50]. To facilitate comparison between our model output and these observations, we extracted temperature and salinity profiles along

an across-fjord transect in the approximate position of an existing conductivity-temperature-depth transect[25,26,49,51] in the middle part of the fjord (location in Fig. 1a), which was obtained within two days of those used as boundary conditions in our simulations. Although we initiated and bounded our model with observations obtained at the fjord mouth, the inclusion of icebergs allows us to better reproduce key aspects of contemporaneous observations made over 60 km up-fjord than in simulations without icebergs (Fig. 7). In particular, the agreement with the cooling observed in the upper ~100–200 m of the domain is greatly improved when icebergs are included, and especially when using the adjusted melt parameters (compare green and grey lines in Fig. 7a).

Although this represents a significant improvement compared to the no-iceberg simulations, there are still differences between the observed and modelled water properties. In particular, the warm spike observed at ~180 m depth, which represents the modified Atlantic Water output (so-called 'glacially modified waters') from the main plume at Helheim Glacier, occurs instead at ~100 m in the iceberg simulations and is cooler than is observed. These differences are perhaps due to the entrainment of additional freshwater from iceberg melting or deflection of plume outflow by the icebergs. In addition, there are a number of relevant parameter values and aspects of the model setup that are poorly constrained by observations; for example, glacier grounding line depth, the rate of entrainment of ambient waters by runoff-driven plumes, the partitioning of runoff along the grounding line, the plume parameterisation used[52] and the effect of suspended sediment on plume dynamics all affect the depth and temperature of the glacially modified waters. Nevertheless, the addition of icebergs represents a marked increase in model realism and substantially improves our ability to model along-fjord changes in water properties compared to previous comparable studies[8,19] and to our no-iceberg simulations (magenta line in Fig. 7).

The freshwater released from icebergs sets up an iceberg melt-driven circulation that is similar to the circulation driven by runoff at the head of glacial fjords. The latter—which for clarity we refer to as the 'runoff-driven circulation' rather than the

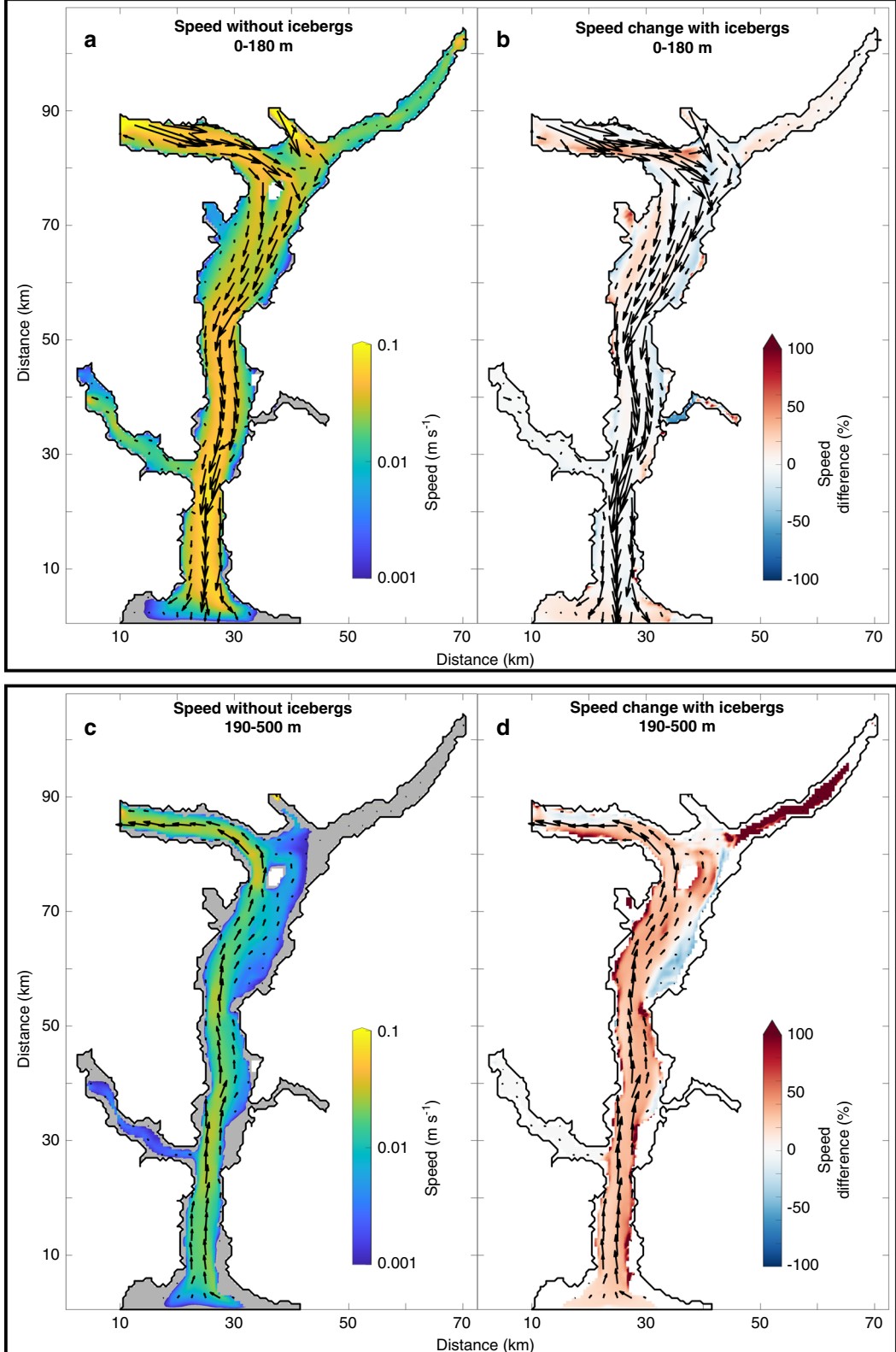

**Fig. 5 The effect of icebergs on fjord circulation in our summer runoff forcing scenario. a** Average current speed between 0 and 180 m depth in the no-iceberg simulation. **b** Relative speed change in the summer runoff forcing scenario compared to the equivalent no-iceberg simulation between 0 and 180 m depth, with positive values indicating an increase in speed. **c, d** As for **a** and **b**, but for the 190–500 m depth band. The arrows indicate the current velocity in the respective simulations averaged over the indicated depths. In all plots, the ocean boundary is at the bottom. Axis distances correspond to those in Fig. 1a.

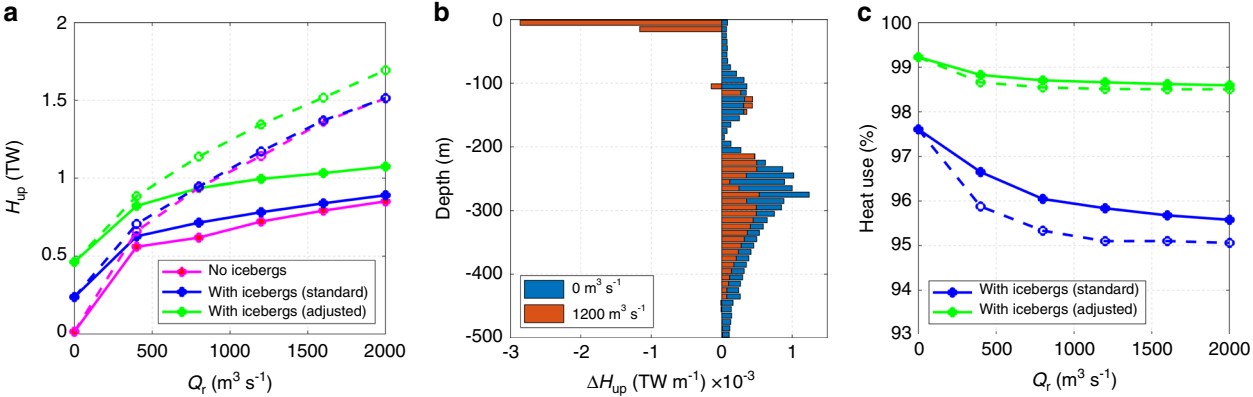

**Fig. 6 The effect of icebergs on along-fjord oceanic heat flux. a** Relationship between runoff ($Q_r$) and up-fjord oceanic heat flux ($H_{up}$) across our flux gate (location in Fig. 1a) in our no-iceberg scenario and in our iceberg scenarios with both the standard and adjusted melt rate parameter values (Methods). **b** Change in up-fjord heat flux ($\Delta H_{up}$) due to icebergs (where a negative value indicates a decrease in heat flux with icebergs at that depth) for the no-runoff (blue) and summer runoff forcing (orange) scenarios. **c** The percentage of heat used for ice melt within the fjord accounted for by iceberg melting. In **a** and **c**, the solid lines represent our 'channelised' drainage scenario and the dashed lines represent our 'distributed' drainage scenario.

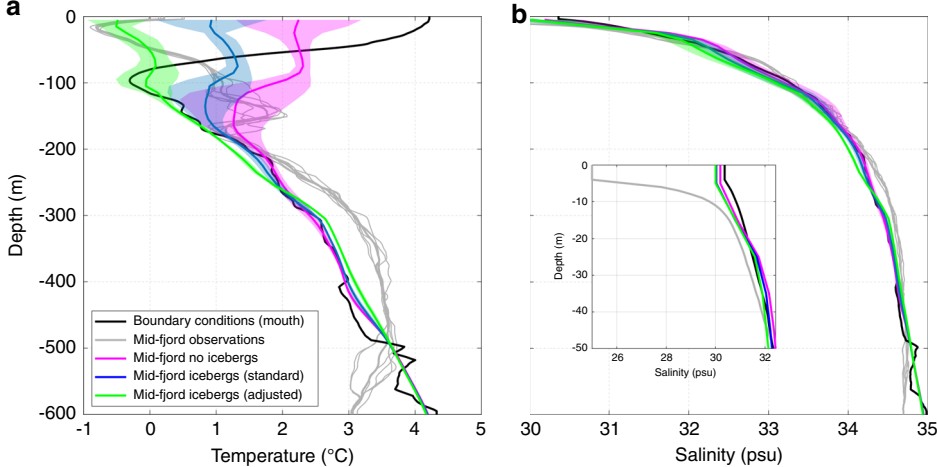

**Fig. 7 Comparison of modelled and observed temperature and salinity.** The magenta lines show modelled temperature and salinity in simulations without icebergs, whilst the blue and green lines show simulations with icebergs using the standard and adjusted melt rate parameter values, respectively (**a** temperature; **b** salinity). The black line shows the average of conductivity-temperature-depth casts at the fjord mouth used as model boundary conditions (locations in Fig. 1a). The grey lines are from casts obtained approximately midway along the fjord (locations in Fig. 1a) roughly contemporaneously to those at the mouth. Modelled mid-fjord profiles were extracted in approximately the same locations as the mid-fjord casts. Shading indicates the standard deviation between simulations with both 'channelised' and 'distributed' drainage configuration scenarios and with runoff varying from 400–2000 m³ s⁻¹, which encompasses the estimated runoff into the fjord (~1000 m³ s⁻¹) at the time the observations were obtained, based on RACMO2.3 modelled runoff. Inset in **b** shows salinity (without shading) in the upper 50 m.

commonly used 'buoyancy-driven circulation'—has received considerable attention in recent years[8,22,25,26,34,53]. The velocity structure of the iceberg melt-driven circulation simulated here is distinct from that of the runoff-driven circulation in several key aspects. Firstly, across-fjord heterogeneity in the velocity structure is diminished compared to the runoff-driven circulation (Fig. 5). Secondly, the fastest down-fjord currents in the runoff-driven circulation are generally simulated close to the head of the fjord (i.e. near the source of buoyancy), whereas in the iceberg melt-driven circulation, the fastest down-fjord currents were generally located much further down-fjord, due to lateral constrictions focusing the flow (Fig. 5). This pattern will, however, likely be sensitive to fjord geometry and bathymetry. Finally, the iceberg melt-driven circulation contains down-fjord currents at slightly greater depths (100–130 m) than the runoff-driven circulation due to some iceberg meltwaters reaching neutral buoyancy deeper in the water column than runoff-driven plumes. The latter

pattern is also implied by inferred submarine meltwater distributions obtained from tracer studies in Sermilik Fjord[23].

Our results suggest that submarine iceberg melting is an important and overlooked driver of fjord circulation, increasing the volume of water exported from the fjord by ~10% in our summer runoff forcing scenario, and can be the dominant driver of fjord circulation when runoff is low. The relative importance of the iceberg melt-driven circulation compared to the runoff-driven circulation during summer will depend on the relative volumes of freshwater derived from icebergs and runoff. For example, in fjords with high rates of iceberg production but comparably low runoff, the iceberg melt-driven circulation may be a key driver of fjord circulation during summer. Similarly, during winter, when runoff is at a minimum, the iceberg melt-driven circulation should act to drive a weak (relative to the runoff-driven circulation) but steady circulation, which may be interrupted by stronger intermediary currents during barrier wind events[18,19]. Since our

modelling suggests that iceberg freshwater flux scales with submerged iceberg surface area at both the individual iceberg-scale and the fjord-scale (Fig. 3c; Supplementary Figs. 3 and 6), as well as with maximum iceberg draught and runoff (Fig. 2b; Supplementary Fig. 7), the iceberg-driven circulation should be a relatively more important driver of fjord circulation in fjords with deeper and more extensive iceberg cover (and greater iceberg melt rates).

Furthermore, our results demonstrate that submarine iceberg melting has important implications for oceanic heat flux towards tidewater glaciers. In simulations with runoff, we simulate an overall ~10–40% increase in up-fjord oceanic heat flux across a flux gate located near Helheim Glacier (flux gate location in Fig. 1a), compared to identical simulations without icebergs (Fig. 6a). Two competing changes to the temperature and velocity structure of the fjords produced these overall changes. The up-fjord volume flux of Atlantic Water increased, thereby increasing the up-fjord heat flux over a broad depth range below ~20 m (Fig. 6b). In contrast, cooling and weakening of up-fjord currents in the upper 20 m caused a reduction in up-fjord heat flux at these depths. The increased up-fjord heat flux below ~20 m implies greater submarine melt-driven undercutting of Helheim Glacier, which can lead to greater iceberg calving rates[12,13]. This result was robust to changes in iceberg size-frequency distribution (Supplementary Table 2) and to wide ranges of iceberg cover (Supplementary Fig. 6) and maximum iceberg draught (Supplementary Fig. 7), with depth-averaged up-fjord heat flux generally increasing above the no-iceberg scenario as either iceberg cover or maximum iceberg draught increase. This suggests that these results may be applicable to many of Greenland's fjords, with the precise impact of icebergs on up-fjord heat flux varying between fjords due to variations in iceberg concentration and keel depth. In addition, these results hint at a potential positive feedback between iceberg production and up-fjord oceanic heat flux, in which greater iceberg production (and therefore freshwater flux) invigorates fjord circulation, leading to an increase in up-fjord oceanic heat flux and therefore calving.

Submarine iceberg melting potentially provides a considerable heat sink in some glacial fjords, but this is difficult to quantify with field-based observations[32,34]. Our modelling suggests that submarine iceberg melting is indeed a large heat sink in Sermilik Fjord, using over 95% of the oceanic heat used for ice melt in our simulations (Fig. 6c). Iceberg melting remained the dominant heat sink under all runoff and drainage scenarios (Methods) and regardless of iceberg draught, aspect ratio or concentration (Supplementary Fig. 9). It is important to emphasise that the heat lost in our simulations is not intended as an accurate representation of the heat budget of Sermilik Fjord because we do not include certain processes (such as atmosphere-ocean interactions, sea ice formation and refreezing, and tidal mixing) that are necessary for calculating the full fjord heat budget[32]. Our modelling does, however, suggest the heat used for submarine iceberg melting is over ten times greater than that used for melting of glacier termini in iceberg-congested fjords like Sermilik Fjord. Whilst we expect that there is considerable uncertainty in this comparison due to, for example, underestimating glacier terminus melt rates in areas distal to runoff plumes[45,46], these results imply that submarine iceberg melting comprises a key component of the fjord heat budget in (at least) iceberg-congested fjords.

Previous field-based investigations have found that the seasonally warm surface layer in glacial fjords has the potential to transport large quantities of oceanic heat towards tidewater glaciers, and causes the majority of the glacial ice-melt[23,32,34,54]. The authors noted that the equivalent terminus melt rates would be unrealistically high if all the heat was used for terminus melting, leading them to suggest that much of the near-surface ocean heat

was likely used to melt icebergs. This interpretation is supported by the results of our model analysis. By implication, a further warming of this layer will expedite iceberg and mélange deterioration, which has in turn been associated with tidewater glacier calving and retreat[6,55].

Several studies have linked either increase in oceanic heat availability[3,4] or increases in up-fjord oceanic heat flux[8] to tidewater glacier retreat. More recently, estimates of ocean thermal forcing during the 21st century have been used to drive parameterisations of glacier retreat as part of the ISMIP6 project[56]. Due to the ice-sheet wide nature and long timescale of this exercise, together with a lack of simple parameterisations for the modification of water masses during fjord transit, the ocean thermal forcing used was based on spatial averages of far-field ocean conditions[57]. We show here that submarine iceberg melting can reduce ocean thermal forcing near the surface, but increase it below, resulting in substantial (~10%) changes in the depth-averaged oceanic heat flux towards tidewater glaciers, with potential implications for glacier submarine melt rates and retreat. Furthermore, our results suggest that a uniform correction applied to ocean conditions at the mouth may not produce an appropriate representation of oceanic heat flux towards tidewater glacier termini because the effect of submarine iceberg melting on up-fjord oceanic heat flux depends on runoff (Fig. 6), as well as on iceberg draught and concentration (Supplementary Figs. 6 and 7), which can vary independently from runoff. Therefore, future studies seeking to examine interactions between the Greenland Ice Sheet and the ocean, over any temporal and spatial scale, should account for iceberg-ocean interactions, particularly when estimating ocean thermal forcing of tidewater glaciers.

## Methods

**Ocean model.** We use the Massachusetts Institute of Technology General Circulation Model[58] (MITgcm), which solves the incompressible Navier-Stokes equations using finite volume methods on an orthogonal curvilinear grid[59]. We take advantage of the non-hydrostatic capability of MITgcm[60] in order to resolve areas of complex bathymetry. MITgcm has been used in numerous studies of ice sheet-ocean interaction in both Greenlandic[8,22] and Antarctic[61] settings.

**Parameterising iceberg melting in MITgcm.** Several parameterisations for bulk iceberg melting exist[62–64], some of which have been incorporated into ocean circulation models. These parameterisations have, for example, proven invaluable for predicting iceberg trajectories and deterioration in the open ocean[64]. To the best of our knowledge, however, these parameterisations have been designed based on iceberg-average submarine melt rates. They would not therefore be suitable tools for simulating vertical variations in iceberg melting within high-resolution domains, such as those required to simulate Greenlandic fjord circulation with high fidelity. We therefore develop a new package to simulate iceberg melting within MITgcm. This package utilises the three-equation melt formulation[47], allowing us to resolve vertical variations in iceberg melt rates, whilst faithfully representing observed iceberg size-frequency and spatial distributions[29,40–42].

**Iceberg geometry.** In Greenland's glacial fjords, icebergs are produced at the fjord head through glacier calving and subsequently drift through the fjord[49], eventually reaching the open ocean or melting out entirely within the fjord. The geometry of the population of icebergs within the fjord can be described in terms of: (1) their size-frequency distribution; (2) their concentration (the fraction of the fjord surface occupied by icebergs); and (3) their aspect ratio (the relationships between their length ($l$), width and keel depth ($d$)).

At a given glacier, iceberg calving events vary in size, typically with many smaller events and relatively few larger events, producing icebergs with dimensions spanning several orders of magnitude[40]. Both field-based[41] and remotely sensed[40] observations show that the resulting iceberg size-frequency distribution can often be approximated using a power law, with the few available observations suggesting that exponents in Greenlandic settings range from $-2.1$ to $-1.8$[40]. In our primary simulations, we generate an array of icebergs with a size-frequency distribution fitting a power law with an exponent of $-2.0$[40], using inverse transform sampling[65] —a classical approach to generating pseudo-random samples from a prescribed probability distribution. We test the sensitivity of our results to this choice of exponent by generating alternative size-frequency distributions with slopes of $-1.8$

and −1.9 (Supplementary Fig. 3; Supplementary Table 2), which encompass the range of size-frequency distributions observed in Sermilik Fjord[31].

In our primary simulations, we based our iceberg setups on a length-draught relationship presented in Barker et al.[42] and on remotely sensed observations of icebergs in Sermilik Fjord presented by Sulak et al.[40] and Enderlin et al.[29,30,48]. The icebergs are rectangular in plan-view and have vertical sides. In our primary simulations, we set the maximum iceberg draught to 300 m[40], and iceberg keel depth was related[42] to iceberg length through $d = 2.91l^{0.71}$. The draught of the resulting icebergs and their distribution are shown in Fig. 1b, c and Supplementary Fig. 1. We also performed secondary simulations with maximum iceberg draughts ranging from 150 to 400 m (Supplementary Fig. 7) and using an alternative relation[40] between iceberg volume ($V$) and plan-view iceberg area ($A$) which states $V = 6.0A^{1\ 30}$ (Supplementary Figs. 3 and 8), and two other iceberg length to keel depth ratios of 2:1 and 1.8:1 (Supplementary Table 2), based on unpublished observations in Sermilik Fjord (see Acknowledgements). In all our simulations, icebergs had length-to-width ratios of 1.62:1[41].

The proportion of the fjord surface covered by icebergs in a given area, $c$, generally decreases towards the fjord mouth. Remotely sensed observations[40] show that $c$ is high (>80%) and uniform throughout the ice mélange, then decreases towards the fjord mouth. Consistent with observations, we used a ~18 km long ice mélange. In our primary simulations, we set $c$ to 80% in the ice mélange and linearly decreased it to 5% near the fjord mouth[40] (Fig. 1c). Slight deviations from this target cover occurred for two reasons. Firstly, icebergs are placed randomly across the width of the fjord. Secondly, these 2-D iceberg distributions must be drawn from a 1-D iceberg size-frequency distribution, resulting in small mismatches between the achieved and target spatial distribution (see Supplementary Text 1 and Supplementary Fig. 1). Using these parameter values, our primary simulations have submerged iceberg surface areas of ~230 km$^2$ in the ice mélange and ~190 km$^2$ in the rest of the fjord. The submerged iceberg areas (Supplementary Table 3) fall within the range of observed values[29]. We also tested the sensitivity of our results to wider ranges of $c$ (Supplementary Fig. 6). This set of sensitivity simulations provides insight into how varying iceberg concentrations over all timescales affects the dynamics of Sermilik Fjord, and provides some indication of the role of iceberg-ocean interaction in the dynamics of other fjords with (typically) lower iceberg concentrations.

In keeping with the implementation of ice shelves[66] and tidewater glaciers[67] in MITgcm, and as in at least one previous representation of iceberg melting within an ocean circulation model[62], we represent icebergs as static entities (i.e. they do not drift or change size over time). Although in reality icebergs drift through the fjord, our representation makes the implementation significantly simpler, and is justified in that our focus is on the impact of iceberg melt on the fjord, rather than the evolution and location of individual icebergs.

**Iceberg thermodynamics**. Icebergs deteriorate due to several processes: melting above and below the waterline due to forced and free convection in air and water, wave erosion, and mechanical breakup[64]. In this study, we consider only submarine melting (due to both forced and free convection below the water line), which is typically the greatest contributor to iceberg freshwater fluxes because of the larger surface area over which it occurs compared to that of the other processes[33].

MITgcm includes representations of freeze-on and melting of both near-horizontal ice shelves[61,66] and vertical ice fronts[67]. We adapt these representations to accommodate our iceberg geometries (i.e. thousands of relatively small vertical and horizontal ice walls scattered throughout the domain). The physics describing ice melting in ocean water remains unchanged and is not described here (see citations above for detailed descriptions of the model physics). Instead, we summarise the key characteristics of our implementation and describe in more detail the changes we have made to better represent submarine iceberg melting.

To calculate submarine iceberg melt rates, we use the velocity-dependent three-equation formulation[47,61,67], in which the rates of heat and salt transfer across the ice-ocean interface are related to the current velocity at the ice-ocean interface through a quadratic drag law[61]. Recent observations[45,46] adjacent to Le Conte Glacier, Alaska, provide strong evidence that typical values used in the three-equation formulation (our 'standard' parameter set) underestimate melt rates of quasi-vertical glacier-ice faces. Based on the Le Conte Glacier observations, Jackson et al.[45] suggested alternative values for three critical parameters in the parameterisation, which together exert a strong control on the rate of heat transfer across the ice-ocean interface. We therefore carried out an additional set of simulations using the adjusted parameter values (Supplementary Table 1) of Jackson et al.[45].

Melt rates derived using the velocity-dependent three-equation formulation are sensitive to the current velocity at the ice-ocean interface. For icebergs, this is the difference between an iceberg's drift velocity and the ambient water velocity at any given point on the iceberg. In ice mélange—a dense matrix of icebergs and sea ice often found adjacent to large tidewater glaciers—iceberg motion is typically slow relative to the surrounding currents[31]; therefore, in this region of our domain, we assume the icebergs are fixed in place. Elsewhere in the domain, we calculate iceberg drift velocity as the average water velocity from the fjord surface to the iceberg keel depth[68,69] (but we do not use this to update the location of each iceberg). We acknowledge that the drift velocity of an iceberg at any time depends on the iceberg's initial drift velocity, wind and water drag, wave action and the

horizontal pressure gradient force exerted by the water, due the displacement of water by the iceberg[64]. However, calculating iceberg drift in this way is computationally intensive and relies on datasets that would be impractical to obtain within MITgcm. We expect this simplification to underestimate the drift velocity (and therefore submarine melt rates) of small icebergs, whose drift is controlled predominately by surface winds[64,70]. The current velocity past each face of a cuboidal iceberg will likely differ substantially; with the lee-side experiencing lower current velocities and the faces oriented parallel to flow experiencing the greatest relative current velocity. To represent this effect, we calculate the submarine melt rate of every face on each iceberg individually at each model vertical level (relative to the calculated drift of the iceberg). We retain vertical profiles of melt rates for each iceberg, but also calculate grid cell-average rates by accounting for the iceberg surface area in each grid cell.

As well as drifting with ocean currents, icebergs also act as a barrier to water flow. We represent this effect using partial cells within MITgcm—essentially forcing a portion of some of the cells to be 'dry'. The fraction of the cell that is dry is equivalent to the proportion of the cell volume occupied by icebergs. In this way, the blocking effect of all of the icebergs in a cell is represented using a single value, rather than representing individual icebergs as solid bodies within grid cells.

The release of meltwater during submarine iceberg melting drives weak buoyant plumes, which can in turn increase melt rates further up the ice face[71]. Resolving such convection-driven melting requires grid cell dimensions that are computationally unfeasible at the fjord scale. Instead, we adopt the approach of Cowton et al.[22] and impose a minimum 'background velocity' at each iceberg face. This value effectively states that there is always some movement of water along the ice face due to melt-driven convection. This approach is similar to another iceberg sidewall melt parameterisation that accounts for melt-driven plume detachment under certain flow regimes[63]. In our primary simulations, we use a value of 0.06 m s$^{-1}$, which is based on a set of simulations utilising the line plume of Jenkins[47] under stratification appropriate for the study fjords[22] and on sparse field-based measurements[35]. To examine the sensitivity of our results to this choice of background velocity, we also tested values of 0.03, 0.09 and 0.12 m s$^{-1}$ in a separate set of simulations (Supplementary Fig. 4). In addition, because the vertical velocity of these plumes theoretically varies through the water column—being greater at a depth where the stratification is weaker—we also conducted sensitivity simulations using a depth-varying background velocity based on two line plume simulations using the initial and the final ambient conditions in our summer runoff forcing scenario (Supplementary Fig. 5; Supplementary Table 2).

**Model setup**

*Model domains.* We generated a model domain representative of Sermilik Fjord in East Greenland (Fig. 1). We used uniform horizontal and vertical resolutions of 500 and 10 m, respectively. The domain extends from Helheim Glacier at the head of the fjord to the area near the mouth where the fjord widens towards the open ocean, and includes all major tributary fjords and glaciers (10 in total). The bathymetry of the domain is based on BedMachine v3[39]. We used a 5 km relaxation zone at the seaward end of the domain, with a relaxation time that increased linearly from 200 s at the open boundary to 5000 s 5 km from the open boundary, to prevent internal reflections of currents created within the domain.

*Initial and boundary conditions.* We utilised conductivity-temperature-depth data (Fig. 1d) obtained in August 2009[25,26,49,51] near the mouth of Sermilik Fjord (locations in Fig. 1a) as the initial conditions. These data are representative of the water masses observed along the adjacent continental shelf during the summer months[25], with warmer, saltier Atlantic Water underlying cooler, fresher Polar Water. In order for the modelled circulation to be the result of only runoff and/or subsurface iceberg melt, the initial potential temperature and salinity conditions were set as horizontally uniform and were kept constant at the boundary throughout each simulation.

*Runoff-driven circulation.* During summer, the primary driver of fjord circulation is usually ice-sheet runoff, which generates buoyant plumes at glacier fronts. These entrain fjord waters before reaching neutral buoyancy (or the fjord surface) and flowing down-fjord[15]. In each simulation, the total runoff entering each fjord was kept constant and was split between each glacier according to the average con-tribution, relative to that of Helheim Glacier, of each respective glacier catchment to the fjord runoff budget during 1990–2012. Runoff from Helheim Glacier was 0–2000 m$^3$ s$^{-1}$, to represent the typical summertime runoff range, based on 1 km$^2$ RACMO2.3 monthly-mean modelled runoff[44]. Glacier catchments (Supplementary Fig. 10) were delineated using standard hydropotential analysis[72] bounded by BedMachine v3[39]. We estimated subglacial discharge at each glacier terminus. To do this, modelled surface runoff[44] was assumed to access the bed immediately (i.e. no supraglacial storage or routing) and was routed to the terminus at 1 m s$^{-1}$ [73,74]. The resulting time-series (1990–2012) of terminus subglacial discharge for each glacier (Supplementary Fig. 10) were temporally-averaged before calculating their contributions to the Sermilik Fjord runoff budget relative to that of Helheim Glacier.

The configuration of the subglacial hydrologic system at the grounding line influences the strength of glacial plumes[75]. The configuration of near-terminus

subglacial hydrological systems remains largely unknown, but potentially ranges from a single channel to fully distributed efflux along the grounding line. Modelling studies have demonstrated that the resulting fjord circulation is sensitive to the strength and distribution of runoff[8,27]. Given this sensitivity, we ran each simulation using two drainage scenarios, using the MITgcm 'iceplume' package[22]. In the first, 'channelised', scenario, 90% of runoff at each glacier entered through a single channel (at the deepest point of the grounding line), while the remaining 10% was divided between smaller channels at 500 m intervals along the grounding line. In the second, 'distributed', scenario, runoff at each glacier was evenly distributed between channels at 500 m intervals.

*Summary of simulation design.* Our simulations (summarised in Supplementary Table 3) are designed to enable evaluation of the impact of submarine iceberg melting on fjord circulation and water properties. As such, we ran one suite of simulations without icebergs (the 'no-iceberg' scenario') and a second suite of simulations with icebergs, which was otherwise identical to the 'no-iceberg' scenario. Within these simulation suites, we performed two sub-suites of simulations to examine the effect of contrasting subglacial hydrological structures. In addition, each of the simulations with icebergs was repeated with adjusted melt rate parameter values[45].

For all of these suites, we ran six simulations for 100 days with runoff from Helheim Glacier varying from 0 to 2000 m$^3$ s$^{-1}$ in increments of 400 m$^3$ s$^{-1}$. Each simulation reached a quasi-steady-state (with domain-averaged kinetic energy changing by <3% over the final 10 model days; Supplementary Fig. 2). This definition of steady-state is based on fjord currents; however, this does not necessarily imply steady temperature and salinity. We therefore also examined modelled time-series of density within the fjord, which show that changes in density by model day 100 were also small (Supplementary Fig. 2). Runoff and open boundary conditions were held constant throughout each simulation. In this way, runoff and submarine iceberg melting are the only forcing in our simulations. We refer to simulations without runoff as 'no-runoff forcing' simulations and note that they are not representative of winter conditions because basal frictional melting of tidewater glaciers during winter likely produces some runoff[76], the conductivity-temperature-depth casts used to create the initial and boundary conditions were obtained in summer (rather than winter) and because we do not simulate the effect of barrier winds and isopycnal heaving, which are common during winter in Sermilik Fjord[18].

*Heat flux.* From a glaciological perspective, whether and how icebergs affect the amount of oceanic heat available to melt tidewater glacier termini is of particular interest. We therefore calculated the oceanic heat transport across a flux gate placed in the ice mélange near Helheim Glacier (location in Fig. 1a). The heat flux, $H$, across each gate was calculated as:

$$H = F_{sea}\rho_0 Q_{gate}\left(\theta_{gate} - \theta_f\right),\tag{1}$$

where $F_{sea}$ is the specific heat capacity of seawater (3980 J kg$^{-1}$ K$^{-1}$), $\rho_0$ is a reference density (1027 kg m$^{-3}$), $Q_{gate}$ is the volume transport across the gate in the direction of interest, $\theta_{gate}$ is the depth-averaged potential temperature across the gate (following Fofonoff and Millard[77]) and $\theta_f$ is the freezing point, calculated as:

$$\theta_f = \lambda_1 S_0 + \lambda_2 + \lambda_3 z,\tag{2}$$

where $\lambda_{1,2,3}$ are the freezing point slope (−0.0573 °C psu$^{-1}$), offset (0.0832 °C) and depth (−0.000761 °C m$^{-1}$), respectively, and $S_0$ is the depth-averaged salinity. To facilitate comparison between the changes in up-fjord heat flux at different depths, and because we are primarily interested in the heat available to melt glacier termini, we use a constant, depth-averaged $\theta_f$ of −2.06 °C based on the average depth of the Helheim Glacier terminus, and the initial salinity stratification[32].

## Data availability

Conductivity-temperature-depth data were requested from the authors of the publication in which the data were originally presented (as cited in text). NASA Operation IceBridge BedMachine Greenland v3 bathymetry and surface data used to generate the model domains and bound our hydropotential analyses are freely available from the National Snow and Ice Data Center (NSIDC) Distributed Active Archive Center via https://doi.org/10.5067/2CIX82HUV88Y. Runoff data used to inform the model forcing are freely available from the corresponding author of Noël et al.[79]. The iceberg size-frequency and distribution data used to inform our model setups are described in Sulak et al.[40]. All of the raw model output data, and the data required to reproduce the simulations, are freely available at the following repository: https://doi.org/10.5281/zenodo.3979647.

## Code availability

The MITgcm code is freely available online via http://mitgcm.org/public/source_code.html. The 'iceplume' package is available upon request from the corresponding author of Cowton et al.[22]. The code required to generate the model input data, the IceBerg model code, and the code required to reproduce the analysis presented here, as well as an example iceberg setup in an idealised fjord, are freely available at the following repository: https://doi.org/10.5281/zenodo.3979647.

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

## Acknowledgements

B.J.D. is funded by a PhD studentship provided by the Scottish Alliance for Geosciences, Environment and Society (SAGES) and the University of St Andrews, UK. We thank F. Straneo, R. H. Jackson, T. Moon and D. A. Sutherland for providing conductivity-temperature-depth data for Sermilik Fjord. We also thank K. M. Schild and D. A. Sutherland for providing guidance on the iceberg aspect ratios in Sermilik Fjord.

## Author contributions

B.J.D. and T.R.C. conceived the study. B.J.D. developed the model code with the support of T.R.C. and A.J.S. B.J.D. designed and conducted the simulations and analysis,

and led the write up of the manuscript. T.R.C., F.R.C. and A.J.S. supported the interpretation of the model results. All authors contributed to the preparation of the manuscript.

## Competing interests

The authors declare no competing interests.
