## [Peer Review File · Nature Communications]

REVIEWER COMMENTS

Reviewer #1 (Remarks to the Author):

Summary

This paper presents a new model development that explicitly includes the input of meltwater from icebergs for the first time in a tidewater glacier fjord, using two sites in southeast Greenland as case studies. The effect of including icebergs is significant, generating substantial currents in the absence of other forcing and modifying fjord water properties (i.e., temperature and salinity) over the upper water column. Overall, these results support the growing notion that icebergs play a critical role in the ice-ocean systems and deserve more study. Indeed, if iceberg melt can influence heat flux in glacier fjords as well as play a role in controlling glacier stability through ice mélange buttressing (not examined here), then in many fjords we might need to reconsider the system as glacier-iceberg-ocean (a term some scientists have started using when considering the effects of mélange already, i.e., ice-mélange-ocean system). However, although the paper is certainly novel and nicely shows the first-order effects of iceberg melt, I believe it is hampered by two main issues: 1) at its heart, this study is a methods paper, as how you parameterize the iceberg melt in the model is key and represents the real novel part of the study (otherwise the scarcity of observations limit the discussion to mostly speculation), and 2) the focus on two fjords muddles the findings and clarity of the paper. To remedy #1, there need to be more details on the iceberg modeling portion and I am not sure that the format of Nature Communications is amenable to that. The community needs this model development and this will be an influential paper. It is not clear, though, that others could repeat these experiments given the information listed. To remedy #2, I would suggest removing the Kangerdlugssuaq Fjord (KF) portion of the manuscript, since you use the iceberg distributions from Sermilik Fjord (SF) for KF and in general there are not large differences in results between the two systems. More interesting to me would be applying your model to a fjord with completely different ice conditions, or watching how Sermilik Fjord transitions from summer to winter (i.e., from when subglacial discharge exists to a time without discharge and ice melt dominates). In summary, I believe this paper represents a significant advance in our capabilities of modeling ice-ocean systems, but want more detail, more clear methods, and more fully discussed results on one application of their novel model developments.

Below, I provide more specific comments going through the paper line by line:

Line 37: add 'and' before 'how those waters...'

Line 53: At what depth? The depth of inflow compensating for these discharge driven circulations is unclear, i.e., is it a small flow over the entire lower layer or is it concentrated below the outflow or above a sill depth?

Line 85: remove 'a' in 'consider a several'

Fig. 2 (and others): these figures would be easier to read/follow if only one fjord was examined here. I know a ton of work went into studying both Sermilik and Kanger, but trying to present everything actually detracts from the punch here.

Line 147-148 and Fig. 3d: This is an inspiring result and plot and what many folks will want, i.e., a simple way to get iceberg freshwater flux from satellite data. The issue here of course is assuming you have good iceberg distributions to begin with, which you don't for KF. Even for SF, there are now papers out (Moyer et al) that look at time variability of iceberg distributions, which could be helpful in deciding how much weight to give these results. To me, this again points at framing this paper more as a proof of concept than a thorough examination of both these systems. There are some nice supplementary figures trying to test the sensitivity of results to certain parameters, such as assumed iceberg shape.

Figure 4: I was very confused by the text describing this figure and couldn't figure out if this was really the 'no runoff' case or the D1 case. In both runoff configurations, I assume the no runoff cases are the same? In any event, more clearly labeling this figure and clear articulation in the text (starting at line 150) is needed here. I think looking at this case is absolutely right, but I wasn't clear what case Fig. 4 was showing.

Line 197: Calculating ocean heat flux, even from models and especially from observations, is always complex. I do think the focus on relative changes is good, i.e., you can't assume all the heat is going into melting and that would change with where you put your flux gate, etc. as you note. However, I wonder if there is a really interesting positive feedback here, in terms of increased iceberg discharge increasing oceanic heat flux leading to more undercutting and enhanced iceberg discharge? You hint at that in the methods (line 528-529), but don't discuss it explicitly.

Line 218-219: Showing that your SF iceberg freshwater fluxes are similar to Moon et al. might not be a very stringent test, given that you are using the same iceberg distributions. Certainly it's a test that your model is doing something reasonable and not generating spurious results, but it might also imply that getting the iceberg distribution right is the most important thing here. You obviously go beyond that study and explore the effects on the ocean circulation and properties- and rightly point out that in SF and KF at least, the effect of iceberg melt in the fjords makes it hard to uniformly adjust far-field conditions a priori.

Fig. 7: Not sure what cumecs is, but I guess $m^3 s^{-1}$?

Line 313: Other authors have noted this as well, including Jackson and Straneo in JPO and Beaird et al.'s work with noble gases, among others.

Line 381: We haven't published it, but we have length to keel depth ratios of 2:1 (i.e., keels are half the longest waterline length) in Sermilik Fjord. This essentially would mean deeper draft icebergs in your study. Happy to share if it helps justify even your sensitivity experiments.

Line 388-389: Doesn't the Moyer et al. study get at the timescale of iceberg distribution steadiness? Can you justify your approach more here?

Line 397: I think wave erosion is the largest contributor to iceberg deterioration, as it leads to mechanical breakup and calving of icebergs from icebergs and the rates can be extremely high. However, you are right that in terms of freshwater flux, the other terms become larger given the subsurface area. Basically, you are assuming the same thing as Moon et al. did in using the Sulak distributions- you have a snapshot in time of icebergs that account for that mechanical breakup.

Line 425: Accounting for relative motion is essential (see some FitzMaurice articles as well as Sutherland et al. 2014 who show motion of icebergs in SF). However, it wasn't clear how you dealt with the mélange area here? In that area, the icebergs are fixed, so all water motion is relative and thus your melt rates should be even higher. Is that what you did or ?

Line 433-436: It is not clear at all what you did with partial cells here. Given a 500 m horizontal resolution, I am still generally confused about how you place individual iceberg geometries into your model grid and still represent the distribution faithfully.

Line 443-446: Moon et al tested different background velocities as well, basing their numbers on the same Jenkins style line plume model initialized with very small runoff. They found the melt rates to be very sensitive to this value, and so readers should be made aware of this. You state tests of other background velocities, but it was unclear if those were shown anywhere in the paper (I don't think so?). Basically, you can increase or decrease the melt rates, and thus the total

freshwater flux, with that knob. Using one constant value is simplest, for sure, but given the stratification present in summer in these fjords, I wonder if it is justified? Also, for strong flows, these ambient 'plumes' might become detached (again see FitzMaurice and others), raising the question of how that affects melt rates along iceberg faces.

Line 544: Can you define what you mean by steady state? In my experience modeling these discharge driven fjords, there is no steady state as the discharge continues to modify the stratification throughout the model run. In fact, that is one reason to consider the effect of icebergs in these models- without that buoyancy forcing, many of these types of simulations drift away from 'reality' over a few weeks of model time.

Methods: You state that your MITgcm setup simulates thousands of tiny vertical and horizontal faces, but don't explicitly mention the modules used to do this- are you using a combined version of the IceFront and ShelfIce packages or both separately or ? Or are the icebergs just bathymetry and you model their thermodynamics another way or some modification of the IcePlume package?

Reviewed by Dave Sutherland

Reviewer #2 (Remarks to the Author):

Review of "Iceberg melting substantially modifies oceanic heat flux towards Greenland's tidewater glaciers" by Davison et al.

This manuscript describes numerical simulations investigating the effect of submarine iceberg melting on fjord circulation and heat transport. Although some of the results of this study are not particularly surprising, i.e. submarine iceberg melting cools and freshens the fjord, the fact that iceberg melting increases the up-fjord heat flux at depth is a novel important result.

In general, I found the manuscript well organized although in several of its part (see below) the authors merely describe the simulations finding without discussing the dynamical mechanisms responsible for those. As such, some revisions are needed to describe the results in more detail and discuss the dynamical mechanisms responsible for the findings of this study. In addition, I suggest the authors to revise the manuscript text and substitute the word 'experiments' with 'simulations', as usually the word 'experiment' is used for a scientific procedure in a laboratory, not using a computer model.

Comments:

1. Figure 1 Caption. Although the authors clarify this in the text, it would be beneficial if it was explicitly said in the figure caption that the iceberg draught-frequency distribution is the one observed in 39.
2. Line 76. The references 35-37 are for Antarctic icebergs which are much larger and may generate a different circulation than the smaller icebergs in Greenland. This possible differences should be acknowledged.
3. Lines 113-115. What is the dynamical mechanism justifying the fact that '... the relationship between iceberg melt rates and runoff is relatively insensitive to iceberg concentration, but is influenced by the spatial pattern of runoff efflux across the grounding line ...'?
4. Lines 119-120. What is the dynamical mechanism justifying this result? Since the icebergs are in the top ~400 m they should be more affected when the plume outflow is trapped in the uppermost layers of the fjord and hence their melting should be sensitive to the increase in runoff.
5. Line 123. I would have expected the range of power 0.1-0.16 to be the same as on line 112, i.e. 0.1-0.2. Is that not the case, i.e. is 0.16 significantly different from 0.2?
6. Line 134. Why are high melt rates found at the sill near the fjord mouth?
7. Figure 3 Caption. How is the 'submerged iceberg surface area' calculated? After reading the method section I had an answer to this question, but a sentence in the main text describing the

geometry of the icebergs used in the model would help the reader interpreting the results described in this figure.

8. Figures 3&4 Caption. I am confused by what the author mean by 'Axis ticks in (a) and (b) correspond to those in Fig 1.' as on Figure 1 there are not axis ticks on panels (a) and (b).
9. Line 141. I suggest clarifying that the 'plume outflow' is near the glacier front.
10. Lines 145-148. Higher melt rate associated with deeply-draughted iceberg are due to the fact that those icebergs are immersed in warmer waters (as discussed on line 142). The increase of submersed iceberg surface area with iceberg drought only holds if the icebergs aspect ratio is approximately the same, i.e. a 'shallow' and 'deep' iceberg could have the same submerged iceberg surface area if their aspect ratio was different. This should be clarified in the text.
11. Line 160-161. I am confused by this sentence because I would not expect the iceberg melt-driven circulation to compete with the runoff circulation, as indeed suggested by Figure 4 (most of the differences in panels (ii) are positive).
12. Figure 4 Caption. Which flow rate was used with the D1 runoff configuration?
13. Figure 5. What does the gray line at 200m represents in panels (c) and (d)?
14. Lines 166-167. What is the dynamical mechanism justifying the increase of up-fjord currents at depth by the iceberg melt-driven circulation?
15. Line 168. The authors should discuss the dynamical mechanism responsible for the increase of submarine iceberg melt rates due to the run-off driven circulation.
16. Line 176. From Figure 5a,c it looks like the inner 80 km are affected, not just the inner 40 km.
17. Line 179. What do the authors mean by 'more homogeneous'? Figure 5b,d shows a decreased cooling confined above the interface between AW and PW waters, not 'homogenous' in the water column.
18. Line 191-193. What is the dynamical mechanism justifying the fact that '... when runoff emerges evenly across tidewater glacier grounding lines, the volume transport across the fluxgate increases...'?
19. Lines 204-205. I am confused by this statement as I thought that near the surface the currents were downs fjord, as shown in Figure 4 (ai) and (ci).
20. Line 259. Since the run-off circulation is mostly down-fjord near the surface, this is not really a characteristic that distinguishes between the iceberg melt-driven circulation and the runoff-driven circulation.
21. Line 264. Lateral constrictions are specific of the two fjords considered in this study. To my knowledge, not all fjords have lateral constriction, so this results is not necessary applicable to all fjords and it should be noted.
22. Line 265-266. Why is this the case? I.e., why does 'the iceberg melt-driven circulation contains down-fjord currents at slightly greater depths than the runoff-driven circulation'?
23. Line 302. See my comment 14. Please explain the dynamical mechanism enhancing the velocity and hence heat flux up-fjord at depth in the presence of icebergs.
24. Line 325-328. I think this point needs a bit of a clarification. If we are interested on how much the glacier face melts, then this finding is important. On the other end, if we are interested in the total freshwater flux or the total melting, i.e. glacier face and icebergs, the ocean thermal forcing based on far-field ocean conditions is fine.
25. Line 383. How is the 'iceberg area A' defined? Surface area, submerged area?
26. Line 408-418. Why was this parameterization for icebergs melting chosen and not, for example, the one suggested in Bigg (2016): The Physics of Icebergs. Would using a different parameterization change the results? The authors may find useful the paper by FitzMaurice and Stern (2018): Parameterizing the basal melt of tabular icebergs, Ocean Modelling, 130, 66-78, where different iceberg melt parameterizations are compared.

Typos:

1. Line 85. The 'a' should be removed.
2. Figure 6 Caption, line 6. The first 'used' should be removed
3. Figure 7, Caption, line 1. 'domai' should read 'domain'.
4. Figure 7, Caption. To be consistent with the rest of the text in the manuscript, I suggest using

m^3/s in place of 'cumsec'.

Reviewer #1 (Remarks to the Author):

Summary

This paper presents a new model development that explicitly includes the input of meltwater from icebergs for the first time in a tidewater glacier fjord, using two sites in southeast Greenland as case studies. The effect of including icebergs is significant, generating substantial currents in the absence of other forcing and modifying fjord water properties (i.e., temperature and salinity) over the upper water column. Overall, these results support the growing notion that icebergs play a critical role in the ice-ocean systems and deserve more study. Indeed, if iceberg melt can influence heat flux in glacier fjords as well as play a role in controlling glacier stability through ice mélange buttressing (not examined here), then in many fjords we might need to reconsider the system as glacier-iceberg-ocean (a term some scientists have started using when considering the effects of mélange already, i.e., ice-mélange-ocean system). However, although the paper is certainly novel and nicely shows the first-order effects of iceberg melt, I believe it is hampered by two main issues: 1) at its heart, this study is a methods paper, as how you parameterize the iceberg melt in the model is key and represents the real novel part of the study (otherwise the scarcity of observations limit the discussion to mostly speculation), and 2) the focus on two fjords muddles the findings and clarity of the paper. To remedy #1, there need to be more details on the iceberg modeling portion and I am not sure that the format of Nature Communications is amenable to that. The community needs this model development and this will be an influential paper. It is not clear, though, that others could repeat these experiments given the information listed. To remedy #2, I would suggest removing the Kangerdlugssuaq Fjord (KF) portion of the manuscript, since you use the iceberg distributions from Sermilik Fjord (SF) for KF and in general there are not large differences in results between the two systems. More interesting to me would be applying your model to a fjord with completely different ice conditions, or watching how Sermilik Fjord transitions from summer to winter (i.e., from when subglacial discharge exists to a time without discharge and ice melt dominates). In summary, I believe this paper represents a significant advance in our capabilities of modeling ice-ocean systems, but want more detail, more clear methods, and more fully discussed results on one application of their novel model developments.

We are grateful to the reviewer for providing a thorough and constructive review of our manuscript. We will start by addressing the two main issues raised, before going through the more specific comments individually.

Firstly, the reviewer has proposed that, given the importance of the novel model development to the paper, the manuscript should contain more methodological detail. We are happy to do this, and in the revised manuscript we have provided further details of the model in both the methods and introduction, whilst retaining the Nature Communications format (which the editor has expressed their support for). With these changes, we believe that there is now sufficient information in the main text for the reader to understand, interpret and critique the results, and enough additional information in the methods for the results to be reproducible.

To further improve transparency and reproducibility, we have also uploaded all of the model code and data required to recreate our simulations, and all raw model output and the analysis code, to an online repository. The repository is available at <http://doi.org/10.5281/zenodo.3979647> and we have linked to it in the code and data availability statement of the manuscript. (If you wish to access the repository, please contact us through Zenodo and use the word “FjordIce” in your access request.) This repository is in a format such that all of the results reported in the manuscript could

be reproduced either from the model output data and analysis code, or (at much greater effort) by re-running all of the simulations using the input data and model code provided. We have also provided a more generic example of the iceberg code, which we hope the readers will be able to use as a starting point for their own investigations. These are described in more detail in a README file within the repository.

We would also argue that, whilst increasing the details on the methods is valuable, we do not wish to detract from the important oceanographic and glaciological implications of the findings presented in our manuscript (which the reviewer notes in their introduction). To increase the visibility, robustness and impact of these findings, we have made some further edits and additions to the revised manuscript. In particular, we now present the results of simulations examining the sensitivity of the key results (e.g. up-fjord heat flux) to melt rate parameter values and a range of iceberg geometries and distributions, which allow improved assessment of the implications of the findings beyond the specific scenarios modelled here. In almost all of these, there is a net increase in depth-averaged up-fjord heat flux. Whilst examining a range of fjord geometries and boundary conditions would be beyond the scope of this manuscript (indeed, we have removed the second fjord geometry, as discussed below), the results of those sensitivity tests indicate that the novel findings on the interaction between iceberg melting and up-fjord heat flux presented in our manuscript are both robust and relevant to many of Greenland's glacier-fjord-iceberg systems. We also believe that there are several other additional findings presented in this manuscript, such as our improved simulation of observed mid-fjord conditions compared to previous studies, which also provide an important contribution to our understanding of Greenland's glacier-fjord-iceberg systems.

Secondly, the reviewer has suggested that we remove Kangerdlugssuaq Fjord from the paper and focus on a single application of our model, which would provide more space to discuss the results from the better-constrained Sermilik Fjord domain and would allow the key findings to be demonstrated and discussed with greater clarity. We have implemented this suggestion and (although painful to set aside the Kangerdlugssuaq work for now), we believe that it is a stronger and clearer manuscript for it.

We would also like to note that in investigating anomalies raised by reviewer #2, we identified a bug in the model code, which affected the modelled temperature and salinity tendencies. We have now fixed this and all of the simulations reported in the revised manuscript have been re-run with the updated code. This has resulted in some changes to the precise values of temperature and salinity within the model runs (by a few tenths of a degree and a few hundredths of a psu on average), but has not changed the overall trends that form the novel results of the study.

Below, I provide more specific comments going through the paper line by line:

Line 37: add 'and' before 'how those waters...'

Done.

Line 53: At what depth? The depth of inflow compensating for these discharge driven circulations is unclear, i.e., is it a small flow over the entire lower layer or is it concentrated below the outflow or above a sill depth?

We now write "*and a compensatory inflowing current over a broad depth range below, typically between the Atlantic Water-Polar Water interface and the sill depth*" (lines 51-54).

Line 85: remove 'a' in 'consider a several'

Done (the whole section has been rephrased).

Fig. 2 (and others): these figures would be easier to read/follow if only one fjord was examined here. I know a ton of work went into studying both Sermilik and Kanger, but trying to present everything actually detracts from the punch here.

Following the reviewer's suggestion, we have removed Kangerdlugssuaq Fjord from the manuscript (see main response above).

Line 147-148 and Fig. 3d: This is an inspiring result and plot and what many folks will want, i.e., a simple way to get iceberg freshwater flux from satellite data. The issue here of course is assuming you have good iceberg distributions to begin with, which you don't for KF. Even for SF, there are now papers out (Moyer et al) that look at time variability of iceberg distributions, which could be helpful in deciding how much weight to give these results. To me, this again points at framing this paper more as a proof of concept than a thorough examination of both these systems. There are some nice supplementary figures trying to test the sensitivity of results to certain parameters, such as assumed iceberg shape.

This is a good point. To remedy this, we have made it clearer in the introduction that we are focusing on a single snap-shot of iceberg conditions – see line 95: *"use a single snapshot-in-time of iceberg cover and distribution"*. We have also conducted several additional simulations with different length-to-keel-depth ratios (using the volume-area relationship from Sulak et al. (2017), the 2:1 ratio that the reviewer mentioned in one of their other comments, and 1.8:1 as a further sensitivity test), because these could affect the relationship shown in the plot. These are all plotted together in Figure 3 of the revised manuscript (with equations of fits shown) and individually in Supplementary Figure 3. We also performed two additional simulations with different iceberg size-frequency distributions (we contacted Dr. Moyer for the relevant data to inform these sensitivity tests) and show some of the results of these in Supplementary Table 2. We chose not to add the output of these simulations to Figure 3 because the effect of the fjord-wide size-frequency distribution on the relationship between submerged surface area and freshwater flux of individual icebergs was minimal.

Figure 4: I was very confused by the text describing this figure and couldn't figure out if this was really the 'no runoff' case or the D1 case. In both runoff configurations, I assume the no runoff cases are the same? In any event, more clearly labeling this figure and clear articulation in the text (starting at line 150) is needed here. I think looking at this case is absolutely right, but I wasn't clear what case Fig. 4 was showing.

We agree that this figure was not adequately described. The figure shows current speeds in simulations with the D1 (now 'channelised') hydrological configuration and $1200 \text{ m}^3 \text{ s}^{-1}$ of runoff. (It is correct that the no runoff cases are the same regardless of drainage configuration, but these are not plotted here). We have also amended both the data plotted in the figure, and the description of the figure in the main text (now lines 159-182), so that their contents are more clearly aligned. The most relevant section is on lines 168-176: *"...some currents in the upper 180 m are slowed by 10-40% because icebergs act as a physical barrier to water flow (Methods) and because in some places the iceberg melt-driven circulation opposes the stronger runoff-driven circulation (Fig 4a). (For example, shallow up-fjord currents, which can be formed in simulations where the plumes reach neutral*

buoyancy below the surface, are opposed by iceberg-melt-driven currents at the same depth). In contrast, in the mélange and near the fjord walls in the upper 180 m, the iceberg melt-driven circulation augments the runoff-driven circulation (Fig. 4a). Throughout the fjord as a whole, up-fjord currents in the 190-500 m depth range are over 30% faster than the equivalent no-iceberg scenario”.

Line 197: Calculating ocean heat flux, even from models and especially from observations, is always complex. I do think the focus on relative changes is good, i.e., you can't assume all the heat is going into melting and that would change with where you put your flux gate, etc. as you note. However, I wonder if there is a really interesting positive feedback here, in terms of increased iceberg discharge increasing oceanic heat flux leading to more undercutting and enhanced iceberg discharge? You hint at that in the methods (line 528-529), but don't discuss it explicitly.

We agree that this is an intriguing possibility, and have added in a short discussion (see lines 319-322) of this potential positive feedback: “...these results hint at a potential positive feedback between iceberg production and up-fjord oceanic heat flux, in which greater iceberg production (and therefore freshwater flux) invigorates fjord circulation, leading to an increase in up-fjord oceanic heat flux and therefore calving”. We are wary though of placing too much emphasis on this because the relationship between melting and calving is complex and the existence of this feedback remains speculative.

Line 218-219: Showing that your SF iceberg freshwater fluxes are similar to Moon et al. might not be a very stringent test, given that you are using the same iceberg distributions. Certainly it's a test that your model is doing something reasonable and not generating spurious results, but it might also imply that getting the iceberg distribution right is the most important thing here. You obviously go beyond that study and explore the effects on the ocean circulation and properties- and rightly point out that in SF and KF at least, the effect of iceberg melt in the fjords makes it hard to uniformly adjust far-field conditions a priori.

This is a good point. In response, we have slightly modified the wording this sentence (lines 234-235) to “...these comparisons give us confidence that our model is realistically capturing iceberg melting within the fjord”. We do still think the comparison is a worthwhile one, as there are few other datasets or publications against which to compare our model and to put some of the results in context. We also think it is a useful opportunity to provide a brief summary of previous comparable estimates of fjord-scale iceberg freshwater fluxes, and so to give credit to previous efforts.

Fig. 7: Not sure what cumecs is, but I guess $m^3 s^{-1}$?

Yes, that is correct. We now use $m^3 s^{-1}$ throughout the manuscript for clarity.

Line 313: Other authors have noted this as well, including Jackson and Straneo in JPO and Beaird et al.'s work with noble gases, among others.

Thank you - we now cite those publications at this point in the revised manuscript (now line 340).

Line 381: We haven't published it, but we have length to keel depth ratios of 2:1 (i.e., keels are half the longest waterline length) in Sermilik Fjord. This essentially would mean deeper draft icebergs in your study. Happy to share if it helps justify even your sensitivity experiments.

We thank the reviewer for sharing this information with us. Since we fix the maximum iceberg draught in our setup to 300 m based on observations in Sulak et al. (2017), using this larger ratio would have the effect of increasing the length of icebergs. Based on this information, we have conducted sensitivity tests using length to keel depth ratios of 2:1 and 1.8:1 (with other parameters identical to our summer runoff forcing scenario). The results of these simulations are most relevant to the relationship between individual iceberg surface area and freshwater flux plotted in Fig. 3c of the main text. We therefore include the results of these simulations in this figure and in Supplementary Figure 3. Overall we find that changing this ratio has only a modest impact on the results (by changing the exponent in the power law relationship between submerged area and freshwater flux by ~ 0.02).

Line 388-389: Doesn't the Moyer et al. study get at the timescale of iceberg distribution steadiness? Can you justify your approach more here?

We thank the reviewer for highlighting this. Yes, looking in particular at Figure S3 of Moyer et al. (2019), there does seem to be some variations in the iceberg distribution month-to-month and between years. Based on this we have made two main changes to the manuscript. Firstly, we have revised the wording of the latter part of the introduction, in which we describe the structure of the investigation, so that we clearly state that iceberg distributions can and do change through time ("*Fjords are dynamic systems, with changes in ... iceberg cover ... occurring over timescales of days to years*" lines 90-91), and that our iceberg distributions are a 'snapshot' in time (line 95) (as the reviewer mentions below). Secondly, we got in touch with Dr. Moyer to discuss this point and to enquire about the availability of the data plotted in their Figure S3. She kindly sent across the data, which we used to inform two additional simulations with different iceberg size-frequency distributions. In Supplementary Table 2, we have provided the effect of using these alternative size-frequency distributions on total iceberg freshwater flux, the vertical partitioning of iceberg freshwater flux and up-fjord heat flux, which we felt were the most important results relevant to these simulations. Changing the size-frequency distribution within the observed range increased the domain freshwater flux from icebergs by less than 3%, increased the freshwater contribution below the pycnocline by 4% and increased up-fjord heat flux by 1.3% compared to our summer runoff forcing scenario.

Line 397: I think wave erosion is the largest contributor to iceberg deterioration, as it leads to mechanical breakup and calving of icebergs from icebergs and the rates can be extremely high. However, you are right that in terms of freshwater flux, the other terms become larger given the subsurface area. Basically, you are assuming the same thing as Moon et al. did in using the Sulak distributions- you have a snapshot in time of icebergs that account for that mechanical breakup.

We thank the reviewer for pointing out this inaccuracy in our wording. In response, we have rephrased this sentence (now lines 434-436) to "*we consider only submarine melting (due to both forced and free convection below the water line), which is typically the greatest contributor to iceberg freshwater fluxes because of the larger surface area over which it occurs compared to that of the other processes*". Earlier in the manuscript (line 95), we clarify that we use a single snapshot of iceberg cover and also note that iceberg distribution can and does change through time.

Line 425: Accounting for relative motion is essential (see some FitzMaurice articles as well as Sutherland et al. 2014 who show motion of icebergs in SF). However, it wasn't clear how you dealt

with the mélange area here? In that area, the icebergs are fixed, so all water motion is relative and thus your melt rates should be even higher. Is that what you did or ?

The model is set up such that the user can provide a 'drift mask', which specifies the cells in which iceberg drift velocity should be calculated. In the mélange area of our domain, we assume the icebergs are fixed (i.e. we assume their drift velocity is zero), so that all water motion is relative. Note that this drift mask is fixed throughout each simulation and the same drift mask was used in all of our primary simulations. We have rephrased this section of the manuscript (lines 454-475) to make our description of this aspect of our model setup clearer. The most relevant section (lines 457-461) now reads: *"In the mélange [] iceberg motion is typically slow relative to the surrounding currents; therefore, in this region of our domain, we assume the icebergs are fixed in place. Elsewhere in the domain, we calculate iceberg drift velocity as the average water velocity from the fjord surface to the iceberg keel depth..."*

Line 433-436: It is not clear at all what you did with partial cells here. Given a 500 m horizontal resolution, I am still generally confused about how you place individual iceberg geometries into your model grid and still represent the distribution faithfully.

The implementation of partial cells is somewhat separate to the thermodynamics routine in the model, and so they are treated slightly differently – in the partial cell implementation, all the icebergs (within a cell) are treated as a single body, whereas in the thermodynamic routine each iceberg (and then each iceberg face) is treated separately. We realise this is not particularly intuitive, however we felt it was an effective means to represent the overall effect that icebergs have on the surrounding water within the functionality available in MITgcm. We have revised our description of our partial cell approach to: *"We represent this effect using partial cells within MITgcm – essentially forcing a portion of some of the cells to be 'dry'. The fraction of the cell that is dry is equivalent to the proportion of the cell volume occupied by icebergs. In this way, the blocking effect of all of the icebergs in a cell is represented using a single value, rather than representing individual icebergs as solid bodies within grid cells".* (now lines 476-481).

More specifically: we calculate the volume of each cell occupied by icebergs, and then use the partial cells functionality in MITgcm to 'close' that portion of the cell (which effectively decreases the vertical thickness of the cell). In that sense, our partial cell approach doesn't 'look' like a collection of icebergs, but instead provides a pragmatic means of approximating the blocking effect that icebergs will have on fjord currents. We believe that it does so without assuming too much understanding of a fairly poorly observed system, and so is preferable to alternative approaches (that we experimented with) such as arbitrarily increasing the viscosity of water in the cell based on the concentration of icebergs.

Line 443-446: Moon et al tested different background velocities as well, basing their numbers on the same Jenkins style line plume model initialized with very small runoff. They found the melt rates to be very sensitive to this value, and so readers should be made aware of this. You state tests of other background velocities, but it was unclear if those were shown anywhere in the paper (I don't think so?). Basically, you can increase or decrease the melt rates, and thus the total freshwater flux, with that knob. Using one constant value is simplest, for sure, but given the stratification present in summer in these fjords, I wonder if it is justified? Also, for strong flows, these ambient 'plumes' might become detached (again see FitzMaurice and others), raising the question of how that affects melt rates along iceberg faces.

This is an important point. The background velocity can and does have a pronounced effect on iceberg freshwater fluxes and we should have included the results of the associated sensitivity tests in the original submission. In the revised version, we highlight these sensitivity tests on lines 235-240: “We note that modelled iceberg melt rates are sensitive to a range of uncertain or temporally variable parameter values, including currents driven by melt-driven convection (Methods; Supplementary Figs. 5 & 6), ... but emphasise that our simulated melt rates are in broad agreement with previous estimates, regardless of these parameter values”. These sensitivity tests are described in more detail in the methods (lines 482-498), with the most relevant part being: “To examine the sensitivity of our results to this choice of background velocity, we also tested values of 0.03, 0.09 and 0.12 m s⁻¹ in a separate set of simulations (Supplementary Fig. 5). In addition, because the vertical velocity of these plumes theoretically varies through the water column – being greater at depth where the stratification is weaker – we also conducted sensitivity simulations using a depth-varying background velocity based on two line plume simulations using the initial and the final ambient conditions in our summer runoff forcing scenario (Supplementary Fig. 5; Supplementary Table 2).”

In an ideal setup, the background velocities would be calculated dynamically using a line plume model (or similar), but for a setup with thousands of icebergs this would represent a significant additional computational expense (and increase in model complexity). Given that previous work (e.g. Cowton et al., 2015; Moon et al., 2017) used a constant background velocity, we think this vertically-varying background velocity is a useful improvement. Given the numerous sources of uncertainty in modelling submarine melt rates of both icebergs and glacier faces, we think it best to present these results at face value, and avoid speculating as to which of these scenarios are more or less accurate.

Line 544: Can you define what you mean by steady state? In my experience modeling these discharge driven fjords, there is no steady state as the discharge continues to modify the stratification throughout the model run. In fact, that is one reason to consider the effect of icebergs in these models- without that buoyancy forcing, many of these types of simulations drift away from ‘reality’ over a few weeks of model time.

We define steady state based on the current velocities in the domain. More specifically, we now clarify on lines 553-556 that: “we ran ... simulations for 100 days” and that “each simulation reached

Figure 1. Example assessment of simulation steady-state conditions from the summer runoff forcing scenario. (a) Domain-averaged kinetic energy. (b) Depth-averaged modelled density through a water column in the middle of the fjord.

a quasi-steady-state (with domain-averaged kinetic energy changing by less than 3% over the final 10 model days; Supplementary Fig. 2)”. As the reviewer implies in their comment, steady state in water properties may never be reached in these kinds of simulations. To assess this, we examined time-series of density in the domain (close to the location of the mid-fjord CTD transect) throughout our simulations. This density time-series varies between simulations, but Figure 1 of this response (and Supplementary Figure 2 of the revised manuscript) shows such a time-series for our summer runoff forcing scenario. Although the density was still varying somewhat by the end of the simulation, the changes in density over time were small. We describe this and cite this figure on lines 557-559: “We therefore also examined modelled time-series of density within the fjord, which show that changes in density by model day 100 were also small (Supplementary Fig. 2)”.

Methods: You state that your MITgcm setup simulates thousands of tiny vertical and horizontal faces, but don't explicitly mention the modules used to do this- are you using a combined version of the IceFront and ShelfIce packages or both separately or ? Or are the icebergs just bathymetry and you model their thermodynamics another way or some modification of the IcePlume package?

The model itself is a new MITgcm package (though it is not yet incorporated in the MITgcm trunk), but it has similarities to iceFront, shelfIce and icePlume. The iceberg faces do not have a physical form (as mentioned above, the 'blocking' effect of icebergs is done separately) – the model just reads and retains information about the location (i.e. which cell) and dimensions (length, width and depth) of each iceberg. In practice, the thermodynamics calculations treat each iceberg face similarly to how iceFront and shelfIce treat vertical and horizontal ice faces, respectively (which we feel is justified since they are all ice walls melting in sea water). The principal difference (from a thermodynamic perspective) between those packages and the iceberg package is that the latter calculates the drift velocity of individual icebergs and the vertical ice faces can be oriented E-W or N-S. From a programming perspective, the challenge was how to effectively deal with what is effectively an irregular matrix with many dimensions (X, Y, number of bergs, berg length, berg width, berg depth) – each modelled horizontal grid cell can contain any number of icebergs, all of different dimensions (and so have their keels in different vertical cells through the water column, typically ending somewhere within a cell rather than at the cell surface or base), each of which must be treated independently and tracked through model time (e.g. so that time-series of melt rates of individual icebergs can be produced). To the best of our knowledge, MITgcm has no in-built function to read and store data of this form, so we developed a means to do this within the iceberg package itself. Unfortunately, this makes the package somewhat slower to run than its contemporaries (iceFront, shelfIce, icePlume). We felt that this level of programming detail would distract from the key scientific points in the paper, but much of this information is included within the code and data repository included in the revised manuscript.

Reviewed by Dave Sutherland

Reviewer #2 (Remarks to the Author):

Review of “Iceberg melting substantially modifies oceanic heat flux towards Greenland’s tidewater glaciers” by Davison et al.

This manuscript describes numerical simulations investigating the effect of submarine iceberg melting on fjord circulation and heat transport. Although some of the results of this study are not particularly surprising, i.e. submarine iceberg melting cools and freshens the fjord, the fact that iceberg melting increases the up-fjord heat flux at depth is a novel important result.

In general, I found the manuscript well organized although in several of its part (see below) the authors merely describe the simulations finding without discussing the dynamical mechanisms responsible for those. As such, some revisions are needed to describe the results in more detail and discuss the dynamical mechanisms responsible for the findings of this study. In addition, I suggest the authors to revise the manuscript text and substitute the word ‘experiments’ with ‘simulations’, as usually the word ‘experiment’ is used for a scientific procedure in a laboratory, not using a computer model.

We are grateful to the reviewer for providing detailed and constructive review of our manuscript. As suggested, in the revised version of the manuscript, we have provided detailed mechanistic explanations for our key results (please see our detailed responses below for more information on these revisions, and for responses to the reviewer’s more specific comments). We also now refer to all computer model runs as simulations, rather than experiments, in the revised manuscript.

In responding to some of the comments raised below, we identified a bug in the model code, which affected the modelled temperature and salinity tendencies. This has now been fixed and all of the simulations reported in the revised manuscript have been re-run with the updated code. This resulted in some changes to the precise values of temperature and salinity within the model runs (by a few tenths of a degree and a few hundredths of a psu on average), but this has not changed the overall patterns that form the novel results of the study.

Comments:

1. Figure 1 Caption. Although the authors clarify this in the text, it would be beneficial if it was explicitly said in the figure caption that the iceberg draught-frequency distribution is the one observed in 39.

We now cite Sulak et al. in the caption of this figure (now reference number 40).

2. Line 76. The references 35-37 are for Antarctic icebergs which are much larger and may generate a different circulation than the smaller icebergs in Greenland. This possible differences should be acknowledged.

We now also cite a similar study (Yankovsky & Yashayaev, 2014) documenting upwelling around two icebergs in the Labrador Sea, which is also consistent with our phrasing in the revised manuscript (i.e. they documented melt-driven upwelling plumes around the icebergs) (see lines 75-77 of the revised manuscript).

Yankovsky, A. E., and Yashayaev, I. 2014. Surface buoyant plumes from melting icebergs in the Labrador Sea. *Deep-Sea Research I*, 91, 1-9.

3. Lines 113-115. What is the dynamical mechanism justifying the fact that ‘... the relationship between iceberg melt rates and runoff is relatively insensitive to iceberg concentration, but is influenced by the spatial pattern of runoff efflux across the grounding line ...’?

We now describe how the relatively warm and fast flowing plume outflow increases iceberg melt rates, particularly for icebergs in the vicinity of glacier fronts. In the ‘distributed’ drainage scenario, we clarify that plume outflow affects a greater proportion of the fjord than in the ‘channelised’ scenario (lines 126-131). In the revised results, we also find a more significant impact of iceberg concentration on iceberg melt rates, which we discuss in the context of the iceberg melt-driven circulation on lines 297-302 and present in Supplementary Figure 6.

4. Lines 119-120. What is the dynamical mechanism justifying this result? Since the icebergs are in the top ~400 m they should be more affected when the plume outflow is trapped in the uppermost layers of the fjord and hence their melting should be sensitive to the increase in runoff.

Following comments from another reviewer, we decided to remove Kangerdluggsuaq Fjord from the manuscript, to improve manuscript structure and allow results from the better constrained Sermilik Fjord to be examined in more detail.

5. Line 123. I would have expected the range of power 0.1-0.16 to be the same as on line 112, i.e. 0.1-0.2. Is that not the case, i.e. is 0.16 significantly different from 0.2?

Thank you for pointing this out - this was due to incorrect weighting of the melt rates by the submerged iceberg surface area within each cell (see comment on correction to the code above). This is fixed in the revised model code and in the revised manuscript. Now, both melt rates and freshwater flux increase equally with runoff (see Figure 2 of the revised manuscript).

6. Line 134. Why are high melt rates found at the sill near the fjord mouth?

This comment related to some of the results from the Kangerdlugssuaq Fjord domain, which we have since removed from the manuscript (see response to comment #4).

7. Figure 3 Caption. How is the ‘submerged iceberg surface area’ calculated? After reading the method section I had an answer to this question, but a sentence in the main text describing the geometry of the icebergs used in the model would help the reader interpreting the results described in this figure.

We now describe our model setup and simulation design in much more detail in the introduction (in general, lines 80-119 are relevant here). Specifically regarding iceberg geometry we write on lines 86-89: “*Individual cuboidal icebergs are roughly oriented with the fjord long-axis and are represented as a set of horizontal and vertical ice faces, with dimensions based on observed iceberg aspect ratios and relationships between iceberg volume and submerged surface area.*” We hope that this (and the surrounding text) is sufficient information for the reader to understand the study design and model

setup (to the extent required to understand and interpret the results) without first reading the methods, where this is described more fully.

8. Figures 3&4 Caption. I am confused by what the author mean by 'Axis ticks in (a) and (b) correspond to those in Fig 1.' as on Figure 1 there are not axis ticks on panels (a) and (b).

We have corrected this to: "*Axis distances in (a) correspond to those in Fig. 1a*" in Figure 3, and "*Axis distances correspond to those in Fig. 1a*" in Figure 4.

9. Line 141. I suggest clarifying that the 'plume outflow' is near the glacier front.

Done (now lines 148-149)

10. Lines 145-148. Higher melt rate associated with deeply-draughted iceberg are due to the fact that those icebergs are immersed in warmer waters (as discussed on line 142). The increase of submersed iceberg surface area with iceberg drought only holds if the icebergs aspect ratio is approximately the same, i.e. a 'shallow' and 'deep' iceberg could have the same submerged iceberg surface area if their aspect ratio was different. This should be clarified in the text.

We now clarify this in the text on lines 149-150: "*...icebergs with greater draughts are more consistently exposed (at least partially) to warmer waters at depth, and so generally melt faster...*", and note that this is "*...assuming the same aspect ratio...*" (Line 155). In addition, we have conducted several sensitivity tests with different iceberg aspect ratios, and find that the relationship between submerged surface area and freshwater flux is similar (Fig. 3c and Supplementary Fig. 3).

11. Line 160-161. I am confused by this sentence because I would not expect the iceberg melt-driven circulation to compete with the runoff circulation, as indeed suggested by Figure 4 (most of the differences in panels (ii) are positive).

Most of the time, the iceberg melt-driven circulation does indeed augment the runoff-driven circulation (as the reviewer points out). This is also the overall effect on the fjord domain (causing a ~10% increase in volume flux in and out of the domain in the revised simulations). However, there are occasions when the iceberg melt-driven circulation opposes the runoff-driven circulation (though the latter is usually stronger than the former). For clarity, we have provided an example of where this is the case (lines 171-173): "*For example, shallow up-fjord currents, which can be formed in simulations where the plumes reach neutral buoyancy below the surface, are opposed by iceberg-melt-driven currents at the same depth*" In most places, and particularly below 180 m, the iceberg melt-driven circulation does of course augment the runoff-driven circulation, resulting in (for example) the net increase in up-fjord volume flux, which is one of the key results reported on in the manuscript. We therefore think that our phrasing "*the iceberg melt-driven circulation and the runoff-driven circulation augment one another when their respective currents are aligned and compete when they are not*" (lines 166-168) is accurate.

12. Figure 4 Caption. Which flow rate was used with the D1 runoff configuration?

$1200 \text{ m}^3 \text{ s}^{-1}$. We now clarify this in the text (line 166) and in the Figure caption. "*Figure 4: The effect of icebergs on fjord circulation using 'channelised' hydrology and $1200 \text{ m}^3 \text{ s}^{-1}$ of runoff...*". Note that we have replaced 'D1' with 'channelised' in the revised manuscript for clarity.

13. Figure 5. What does the gray line at 200m represents in panels (c) and (d)?

This is an artefact of the way we have created the plots and is actually just a boundary between two axes, so does not represent anything meaningful. We have removed this boundary in the revised figures.

14. Lines 166-167. What is the dynamical mechanism justifying the increase of up-fjord currents at depth by the iceberg melt-driven circulation?

We have clarified (on lines 164-165) that this is a compensatory current for the melt-driven surface outflow.

15. Line 168. The authors should discuss the dynamical mechanism responsible for the increase of submarine iceberg melt rates due to the run-off driven circulation.

We now discuss (on lines 126-131) how the relatively warm and fast plume outflow increases heat transfer across the iceberg-ocean interface, particularly for icebergs in the vicinity of glacier fronts (where plume outflow is strongest).

16. Line 176. From Figure 5a,c it looks like the inner 80 km are affected, not just the inner 40 km.

Fixed. We have rephrased this to: “...we simulate iceberg-induced cooling and freshening of up to 5°C and 0.7 psu throughout the upper ~100 m, though the changes are most pronounced near the fjord head and surface (Fig. 5b)” (see lines 184-186).

17. Line 179. What do the authors mean by 'more homogeneous'? Figure 5b,d shows a decreased cooling confined above the interface between AW and PW waters, not 'homogenous' in the water column.

We acknowledge that our original wording was not clear. We now write (see lines 186-188): “With the addition of runoff, the invigorated circulation results in more uniform iceberg-induced cooling of ~1°C and freshening of ~0.1 psu above the Atlantic Water-Polar Water interface...”, which we think is a more accurate description of the simulated pattern of changes to the temperature and salinity structure of the fjord in this scenario.

18. Line 191-193. What is the dynamical mechanism justifying the fact that ‘... when runoff emerges evenly across tidewater glacier grounding lines, the volume transport across the fluxgate increases...’?

The strength of the runoff-driven circulation is governed primarily by plume entrainment. As shown in several publications (e.g. Jenkins, 2011; Cowton et al., 2016), for an individual plume, entrainment increases sub-linearly with runoff. Therefore, for a given runoff, the total entrainment (and therefore the total plume volume flux) is greater if there are multiple smaller plumes, rather than concentrating that runoff into one large plume. In the revised version of the manuscript, we have explained this briefly (on lines 200-202) in the main text as “... because of more efficient entrainment of ambient waters into glacial plumes per unit volume of runoff [in the ‘distributed’ scenario], due to the sub-linear relationship between entrainment of ambient waters into plumes and runoff”.

Jenkins, A. 2011. Convection-driven melting near the grounding lines of ice shelves and tidewater glaciers. *Journal of Physical Oceanography*, 41, 2279-2294.

Cowton, T., Sole, A., Nienow, P., Slater, D., Wilton, D., and Hanna, E. 2016. Controls on the transport of oceanic heat to Kangerdlugssuaq Glacier, East Greenland. *Journal of Glaciology*.

19. Lines 204-205. I am confused by this statement as I thought that near the surface the currents were down fjord, as shown in Figure 4 (ai) and (ci).

In general, this is true. However, in the mélange area near Helheim Glacier, we simulated some up-fjord currents in the upper 20 m (which are responsible for the limited up-fjord heat transport at those depths). These currents are visible as a thin red band in the upper right hand portion of Fig. 5f. We have clarified this in the manuscript as “...because of cooling and weakening of up-fjord surface currents found in the mélange (Fig. 5f) and strengthening of up-fjord currents below” (lines 213-214).

20. Line 259. Since the run-off circulation is mostly down-fjord near the surface, this is not really a characteristic that distinguishes between the iceberg melt-driven circulation and the runoff-driven circulation.

Much of the time, this is correct, and so we have removed this point as a distinguishing feature of the iceberg melt-driven circulation (this section is now lines 271-286).

21. Line 264. Lateral constrictions are specific of the two fjords considered in this study. To my knowledge, not all fjords have lateral constriction, so this results is not necessary applicable to all fjords and it should be noted.

This is a good point. We now add (line 281-282): “*This pattern will, however, likely be sensitive to fjord geometry and bathymetry*”.

22. Line 265-266. Why is this the case? I.e., why does ‘the iceberg melt-driven circulation contains down-fjord currents at slightly greater depths than the runoff-driven circulation’?

This is because some freshwater originating from iceberg melting reaches neutral buoyancy at greater depth than the plume outflow. We explain this on lines 282-285: “*the iceberg melt-driven circulation contains down-fjord currents at slightly greater depths (100-130 m) than the runoff-driven circulation due to some iceberg meltwater reaching neutral buoyancy lower in the water column than runoff-driven plumes*”.

23. Line 302. See my comment 14. Please explain the dynamical mechanism enhancing the velocity and hence heat flux up-fjord at depth in the presence of icebergs.

We have clarified (on lines 164-165) that this is a compensatory current for the relatively fresh and cold iceberg melt-driven outflow above.

24. Line 325-328. I think this point needs a bit of a clarification. If we are interested on how much the glacier face melts, then this finding is important. On the other end, if we are interested in the

total freshwater flux or the total melting, i.e. glacier face and icebergs, the ocean thermal forcing based on far-field ocean conditions is fine.

We have added “...with potential implications for glacier submarine melt rates and retreat” (lines 354-355), which seems appropriate given that the context of the paragraph is glacier retreat and the component of the ISMIP6 project that sought to estimate glacier retreat and submarine melting from ocean thermal forcing.

25. Line 383. How is the 'iceberg area A' defined? Surface area, submerged area?

This is the area of the upper surface of the iceberg in plan-view (clarified on line 404).

26. Line 408-418. Why was this parameterization for icebergs melting chosen and not, for example, the one suggested in Bigg (2016): The Physics of Icebergs. Would using a different parameterization change the results? The authors may find useful the paper by FitzMaurice and Stern (2018): Parameterizing the basal melt of tabular icebergs, Ocean Modelling, 130, 66-78, where different iceberg melt parameterizations are compared.

We thank the reviewer for directing us to the very useful and interesting paper (now cited on line 427). We chose to use the three-equation formulation and not one of the other parameterizations for iceberg melting because the other parameterisations for iceberg melting were (to the best of our knowledge) designed based on iceberg-averaged submarine melt rates, and so are not best suited for resolving the vertical variations in iceberg melt rates, which we thought would be crucial for accurately quantifying the impact of submarine iceberg melting on fjord circulation. The three-equation formulation has been used many times in similar (Greenlandic) settings, including both vertical and horizontal ice faces, so we felt it was well-suited to this application. Furthermore, the use of the three-equation formulation maintains consistency with packages within MITgcm for simulating melt of glaciers and ice shelves (and which we use to calculate the submarine melting of glacier termini in our study).

Typos:

1. Line 85. The 'a' should be removed. *Corrected.*
2. Figure 6 Caption, line 6. The first 'used' should be removed *Done.*
3. Figure 7, Caption, line 1. 'domai' should read 'domain'. *Fixed.*
4. Figure 7, Caption. To be consistent with the rest of the text in the manuscript, I suggest using m^3/s in place of 'cumsec'. *Done.*

REVIEWERS' COMMENTS

Reviewer #1 (Remarks to the Author):

Dear authors and editor,

I am impressed by the response to the previous (original) review and think the manuscript is overall much clearer than before. I realize how hard it is to take work out of a manuscript (the Kanger case study), but I do think it helps in this case. I am satisfied my main comments have been addressed. I have only a few remaining minor comments below:

Line 132-133: It's not clear where these integrated values of iceberg melt come from. From my eye, I can't get to the higher values from Fig 2b...

Line 138: I don't see a Fig. 2c?

Line 160: Fig. 4 does not show a no runoff case, so citing it here was confusing to me.

Line 184-185: It might be interesting to note how close these changes in T/S are to offsetting each other, that is, I think at these temperatures a 1psu change is near a 5C change? In any event, it's probably worth mentioning how these gradients in T/S might or might not show up as density gradients, since those are what are dynamically important. That is, can you isolate the thermal and haline effects on density?

Line 206: Am I reading Fig. 6a right that there's not a big difference in heat transport in the distributed case if you add icebergs (above a certain Q_r)? I didn't see this mentioned in the text at all.

Line 244: another instance where it'd be good to compare with the resulting density gradients, if any.

Line 267: I appreciate the revisions the authors have made to make their methods clearer. This is one additional instance where they could be more precise. That is, can you be more explicit by what you mean here in terms of substantial model improvement? In terms of skill at representing T profile? What about S? Or other variables? Is it good at getting the trends but not the magnitude? I don't want this to turn into a modeling paper, but if you claim substantial improvement in modeling performance, you should be explicit about what you mean.

Line 260 and throughout: I wonder if the missing 'warm' anomaly at 100 m depth could also be fixed by tuning the subglacial discharge and playing with the way you parameterize that, a la Jackson et al. I think you could get the Helheim (and maybe other glaciers' discharge) plumes more correct if you wanted to. But, obviously, that's not the point of this paper, so I am not asking you to do it, just pointing out that you could justify that gap.

Line 407: Use 1.62:1 here to be consistent with other ratios.

Line 518: In the main text you explicitly don't use the term 'buoyancy-driven', so why have it here?

Supplemental

First paragraph: Note that the Sulak et al. data are available online, if you ever want to calculate those distributions yourself.

Supp Fig. 8: Instead of using Barker and Sulak, is it easier to state the aspect ratios? I think this

comment stems from a little confusion still about the differences between the 1.8:1, 2:1, Sulak, and Barker runs you did. Some change aspect ratio and some just change volume to area? Somehow this needs to be made clearer.

Supp. Table 3: Is it obvious why the 1.8:1 ratio case has higher submerged area values than the 2:1 case? Again, this is another place where the differences between Sulak/Barker/2:1/1.8:1 could be made more explicit.

Reviewer #2 (Remarks to the Author):

Review of "Iceberg melting substantially modifies oceanic heat flux towards Greenland's tidewater glaciers" by Davison et al.

In their response, the authors have provided convincing arguments to my comments and have discussed the dynamical mechanisms responsible for the simulations finding. I agree that removing the results for Kangerdluggsuaq Fjord from the manuscript improved its structure and allowed a more detail discussion of the results for Sermilik Fjord. In couple of instances, see below, I think the reader would benefit from some extra details, but beside these very minor points I'm in favor of publishing this manuscript in Nature Communications.

Comments:

1. (Previous point 14: Lines 166-167. What is the dynamical mechanism justifying the increase of up-fjord currents at depth by the iceberg melt-driven circulation?) On lines 164-165 the authors describe the dynamical mechanism justifying the increase of up-fjord currents at depth by the iceberg melt-driven circulation as a 'compensatory current', but it is still unclear what they mean by 'compensatory'. Are they thinking of conserving volume? Or is the entrainment in the melt-driven surface outflow current driving the up-fjord currents at depth, i.e. like the classic estuarine flow? A sentence clarifying this point would be beneficial to the reader.

2. (Previous point 26: Line 408-418. Why was this parameterization for icebergs melting chosen and not, for example, the one suggested in Bigg (2016): The Physics of Icebergs. Would using a different parameterization change the results? The authors may find useful the paper by FitzMaurice and Stern (2018): Parameterizing the basal melt of tabular icebergs, Ocean Modelling, 130, 66-78, where different iceberg melt parameterizations are compared.) I think the reader would benefit from a discussion in the text justifying the choice of this particular parameterization for icebergs. I accept the justification given in their rebuttal, but I think it should be included in the manuscript.

Reviewer #1 (Remarks to the Author):

Dear authors and editor,

I am impressed by the response to the previous (original) review and think the manuscript is overall much clearer than before. I realize how hard it is to take work out of a manuscript (the Kanger case study), but I do think it helps in this case. I am satisfied my main comments have been addressed. I have only a few remaining minor comments below:

Line 132-133: It's not clear where these integrated values of iceberg melt come from. From my eye, I can't get to the higher values from Fig 2b...

This statement should have referred to Figure 2a (see the right y-axis), which does show the higher freshwater flux values. Fig 2b shows only the channelised hydrology and standard melt rate parameter value cases, so would not produce the higher freshwater fluxes shown by the green lines in Fig 2a – we have clarified this in the Figure caption: “(b) Horizontally-averaged iceberg freshwater flux profile, coloured by runoff, for the ‘channelised’ runoff configuration and standard melt rate parameter values”.

Line 138: I don't see a Fig. 2c?

This should have referred to Fig 2b, and has been corrected in the revised manuscript. (Also corrected on lines 134 and 141).

Line 160: Fig. 4 does not show a no runoff case, so citing it here was confusing to me.

We now only cite Fig. 5 of the previous submission. To ensure Figures are provided in the order in which they are cited, we have switched the order of Figures 4 and 5 in the revised manuscript.

Line 184-185: It might be interesting to note how close these changes in T/S are to offsetting each other, that is, I think at these temperatures a 1psu change is near a 5C change? In any event, it's probably worth mentioning how these gradients in T/S might or might not show up as density gradients, since those are what are dynamically important. That is, can you isolate the thermal and haline effects on density?

In the revised version, we have added a description of this in our results (lines 193-196): “The freshening and cooling have compensating effects on the fjord water density. Thus, the net effect on density from iceberg melt is small (typically much less than 1% change in any location, compared to the no-iceberg scenario) and the resulting lateral density gradients within the fjord are weak.” It would certainly be possible to isolate the thermal and haline effects on density, but we felt that this was not necessary given the changes in density are modest.

Line 206: Am I reading Fig. 6a right that there's not a big difference in heat transport in the distributed case if you add icebergs (above a certain Q_r)? I didn't see this mentioned in the text at all.

Yes, that is correct. We now describe this in the text on lines 214-216: “This overall increase in up-fjord heat flux is most pronounced with low values of runoff and with ‘channelised’ hydrology (Fig. 6a), when the iceberg melt-driven circulation is a relatively more important driver of fjord circulation.”

Line 244: another instance where it'd be good to compare with the resulting density gradients, if any.

There are weak along-fjord density gradients, so we now write: "The impact on water column temperature and salinity increases towards the fjord head, where iceberg concentrations are greatest, resulting in along-fjord gradients in temperature, salinity and density." (Lines 250-252).

Line 267: I appreciate the revisions the authors have made to make their methods clearer. This is one additional instance where they could be more precise. That is, can you be more explicit by what you mean here in terms of substantial model improvement? In terms of skill at representing T profile? What about S? Or other variables? Is it good at getting the trends but not the magnitude? I don't want this to turn into a modeling paper, but if you claim substantial improvement in modeling performance, you should be explicit about what you mean.

We agree that it is important to be explicit here. We now write: "Nevertheless, the addition of icebergs represents a marked increase in model realism and substantially improves our ability to model along-fjord changes in water properties compared to previous comparable studies^{8,19} and to our no-iceberg simulations" (see lines 275-278).

Line 260 and throughout: I wonder if the missing 'warm' anomaly at 100 m depth could also be fixed by tuning the subglacial discharge and playing with the way you parameterize that, a la Jackson et al. I think you could get the Helheim (and maybe other glaciers' discharge) plumes more correct if you wanted to. But, obviously, that's not the point of this paper, so I am not asking you to do it, just pointing out that you could justify that gap.

We agree that the magnitude of the subglacial discharge and the way the plume is parameterized could explain this difference. Figure 7 already includes a large range of subglacial discharge values, so we don't think it would make sense to list this here as a reason for the model-observation mismatch. However, we do now include the way the plume is parameterized as a further reason, and cite Jackson et al. (2017). (See line 273).

Line 407: Use 1.62:1 here to be consistent with other ratios.

Done.

Line 518: In the main text you explicitly don't use the term 'buoyancy-driven', so why have it here?

We have changed this to 'Runoff-driven circulation' to be consistent with the rest of the text.

Supplemental

First paragraph: Note that the Sulak et al. data are available online, if you ever want to calculate those distributions yourself.

We thank the reviewer for alerting us to this. We have removed the majority of this paragraph and have instead just listed the inputs provided for the model and noted (with a citation to Sulak et al.) that these inputs are observable.

Supp Fig. 8: Instead of using Barker and Sulak, is it easier to state the aspect ratios? I think this

comment stems from a little confusion still about the differences between the 1.8:1, 2:1, Sulak, and Barker runs you did. Some change aspect ratio and some just change volume to area? Somehow this needs to be made clearer.

Yes, all of them (except the 'Sulak' experiments), are based on prescribed aspect ratios. The 'Sulak' experiments were based on a volume-area relationship. However, since we also set the maximum iceberg draught and the shape of the icebergs, the latter also results in a change to the iceberg aspect ratio. We have updated the legend in this figure, as well as in Supplementary Figure 9 and the caption of Figure 3 in the main text, to reflect this and to maintain consistency throughout the manuscript.

Supp. Table 3: Is it obvious why the 1.8:1 ratio case has higher submerged area values than the 2:1 case? Again, this is another place where the differences between Sulak/Barker/2:1/1.8:1 could be made more explicit.

Yes, this is because the shorter length of the icebergs in the 1.8:1 necessitates a greater number of icebergs in order to reach the target concentration, ultimately increasing the submerged iceberg area. To make the differences between the experiments clearer, we have updated the description of the experiments in this table to be consistent with the description in the methods and figures mentioned above.

Reviewer #2 (Remarks to the Author):

Review of “Iceberg melting substantially modifies oceanic heat flux towards Greenland’s tidewater glaciers” by Davison et al.

In their response, the authors have provided convincing arguments to my comments and have discussed the dynamical mechanisms responsible for the simulations finding. I agree that removing the results for Kangerdluggsuaq Fjord from the manuscript improved its structure and allowed a more detail discussion of the results for Sermilik Fjord. In couple of instances, see below, I think the reader would benefit from some extra details, but beside these very minor points I’m in favor of publishing this manuscript in Nature Communications.

Comments:

1. (Previous point 14: Lines 166-167. What is the dynamical mechanism justifying the increase of up-fjord currents at depth by the iceberg melt-driven circulation?) On lines 164-165 the authors describe the dynamical mechanism justifying the increase of up-fjord currents at depth by the iceberg melt-driven circulation as a ‘compensatory current’, but it is still unclear what they mean by ‘compensatory’. Are they thinking of conserving volume? Or is the entrainment in the melt-driven surface outflow current driving the up-fjord currents at depth, i.e. like the classic estuarine flow? A sentence clarifying this point would be beneficial to the reader.

By ‘compensatory’, we were indeed thinking of conserving volume, and so the fjord water entrained in the melt-driven surface outflow is also important. We now clarify this on lines 165-167: “This weak but broad up-fjord current compensates (in terms of volume) for the fjord water entrained in the relatively fresh and cold iceberg melt-driven outflow above”.

2. (Previous point 26: Line 408-418. Why was this parameterization for icebergs melting chosen and not, for example, the one suggested in Bigg (2016): The Physics of Icebergs. Would using a different parameterization change the results? The authors may find useful the paper by FitzMaurice and Stern (2018): Parameterizing the basal melt of tabular icebergs, Ocean Modelling, 130, 66-78, where different iceberg melt parameterizations are compared.) I think the reader would benefit from a discussion in the text justifying the choice of this particular parameterization for icebergs. I accept the justification given in their rebuttal, but I think it should be included in the manuscript.

We agree that the reader would benefit from a discussion in the text justifying the choice of melt parameterisation. We have therefore added in a paragraph in our methods section (lines 387-396) justifying this choice: “Several parameterisations for bulk iceberg melting exist⁶¹⁻⁶³, some of which have been incorporated into ocean circulation models. These parameterisations have, for example, proven invaluable for predicting iceberg trajectories and deterioration in the open ocean⁶³. To the best of our knowledge, however, these parameterisations have been designed based on iceberg-average submarine melt rates. They would not therefore be suitable tools for simulating vertical variations in iceberg melting within high-resolution domains, such as those required to simulate Greenlandic fjord circulation with high fidelity. We therefore develop a new package to simulate iceberg melting within MITgcm. This package utilises the three-equation melt formulation⁴⁷, allowing us to resolve vertical variations in iceberg melt rates, whilst faithfully representing observed iceberg size-frequency and spatial distributions^{29,40-42}.”